# Diversity-dependent speciation and extinction in hominins

**Laura A. van Holstein** ✉ **& Robert A. Foley**

The search for drivers of hominin speciation and extinction has tended to focus on the impact of climate change. Far less attention has been paid to the role of interspecific competition. However, research across vertebrates more broadly has shown that both processes are often correlated with species diversity, suggesting an important role for interspecific competition. Here we ask whether hominin speciation and extinction conform to the expected patterns of negative and positive diversity dependence, respectively. We estimate speciation and extinction rates from fossil occurrence data with preservation variability priors in a validated Bayesian framework and test whether these rates are correlated with species diversity. We supplement these analyses with calculations of speciation rate across a phylogeny, again testing whether these are correlated with diversity. Our results are consistent with clade-wide diversity limits that governed speciation in hominins overall but that were not quite reached by the *Australopithecus* and *Paranthropus* subclade before its extinction. Extinction was not correlated with species diversity within the *Australopithecus* and *Paranthropus* subclade or within hominins overall; this is concordant with climate playing a greater part in hominin extinction than speciation. By contrast, *Homo* is characterized by positively diversity-dependent speciation and negatively diversity-dependent extinction—both exceedingly rare patterns across all forms of life. The genus *Homo* expands the set of reported associations between diversity and macroevolution in vertebrates, underscoring that the relationship between diversity and macroevolution is complex. These results indicate an important, previously underappreciated and comparatively unusual role of biotic interactions in *Homo* macroevolution, and speciation in particular. The unusual and unexpected patterns of diversity dependence in *Homo* speciation and extinction may be a consequence of repeated *Homo* range expansions driven by interspecific competition and made possible by recurrent innovations in ecological strategies. Exploring how hominin macroevolution fits into the general vertebrate macroevolutionary landscape has the potential to offer new perspectives on longstanding questions in vertebrate evolution and shed new light on evolutionary processes within our own lineage.

Leverhulme Centre for Human Evolutionary Studies, Department of Archaeology, University of Cambridge, Cambridge, UK. ✉e-mail: lav22@cam.ac.uk

The diversification of a lineage is the net output of speciation minus extinction. A theme that runs through much research into human evolution is whether the determinants of hominin diversification conform to or diverge from those seen in other taxa. At one extreme lie ideas such as Wolpoff's 'single species hypothesis'[1], which suggested that there can be no speciation in the hominin lineage, as its niche is 'culture'. Culture, in Wolpoff's view, is uniquely human and prevents boundaries between populations from occurring; hence, speciation was prohibited in hominins, but not in other clades. At the other extreme are interpretations that emphasize commonalities between patterns of hominin speciation and extinction and those of other clades[2]. Within this group, research interest has primarily been devoted to examining the role of climate in shaping hominin diversification[2–7]. What has received far less attention as a potential driver of hominin diversification than climate, however, is competition.

Competition occurs across taxonomic scales, from interindividual competition within populations[8] to intergroup competition within species[9] and interspecific competition[10]. Competition at each of these levels has been shown to act as an important driver of evolution at equal or higher scales[11–13]. Here we focus on interspecific competition for niches (hereafter 'competition') and its consequences above the species level. Although the concept of 'niche' has only rarely been formally defined in previous work on diversity-dependent speciation[14–16], its implicit definition in previous work is that of a Hutchinsonian ecological niche[17]—an $n$-dimensional hypervolume describing all environmental resources and conditions required for species persistence. We adopt this conventional definition throughout this paper. The consequences of competition can include three processes: speciation, extinction and morphological change through, for example, character displacement[13,18]. There is some indirect evidence that competition resulted in morphological evolution in our lineage: competition between *Homo* and *Paranthropus* in East Africa probably led to character displacement in the mandibular premolar morphology of these two groups[19]. However, much less work has been devoted to exploring the effects of competition on hominin speciation or extinction.

Ecological competition with large carnivores is thought to have exerted a strong effect on hominin ranging patterns[20,21], hunting behaviour[22] and, of particular interest at a macroevolutionary scale, geographic dispersals[23,24]. Although it is unknown whether competition with large carnivores had direct effects on hominin speciation or extinction, the link between dispersals and these macroevolutionary processes is well established[25]. Compared with competition between hominins and non-hominins, the dynamics and effects of competition between hominin species have received comparatively little attention. Although competition may have contributed to a pulse of hominin extinction around 1.5 million years ago[26], and some recent reviews have used evidence for hominin sympatry in East and South Africa to suggest the possibility of competition[27,28], an explicit investigation of the extent to which competition drove hominin diversification is lacking.

Competition has probably had a major role in animal diversification, however, leaving signals in correlations between species diversity, on the one hand, and speciation and extinction on the other[14,16,29–31]. Speciation can be both positively or negatively diversity dependent or occur independently of a clade's own diversity. Under positive diversity dependence, speciation rates rise as a function of the novel evolutionary opportunities and interactions created by other species[32]. This pattern is exceptionally rare among all forms of life, however, having been reported only in island-dwelling beetles[33], plants and arthropods[34], and this latter case is contentious[35,36]. Instead, if a relationship exists between vertebrate speciation and diversity, this is usually negative[14,16,30,31]. There are two processes by which speciation may be negatively controlled by diversity: competition for (1) niche space, or (2) geographic space[15]. In both, speciation is regulated by bounded ecological opportunities. In classical Darwinian diversity dependence[16,37], speciation into a niche occupied by a closely related

species is prohibited, producing a negative relationship. At a higher level of taxonomic organization, models of asymptotic diversity predict slowdowns in speciation as a finite number of niches within an adaptive grade, or a finite number of ranges within bounded space, become occupied by closely related species as a clade grows[25,38]. However, findings of diversity-independent speciation in some clades has led to intense debate about whether negative diversity dependence is universal across vertebrates; the same is true for the related question of whether absolute limits to niches or geographic ranges even exist[39,40].

The relationship between extinction and diversity has received less explicit empirical attention than that between speciation and diversity. However, when a relationship is reported, extinction is typically positively diversity dependent[41,42]. These patterns align with expectations based on theory. Under Darwinian diversity dependence, competition between ecologically similar species should result in extinction of outcompeted species[37] even in the absence of absolute limits to species diversity. Models of asymptotic diversity[16,43] predict increased rates of extinction as species diversity approaches an explicitly predicted diversity limit. Asymptotic diversity dynamics have been reported for multiple vertebrate clades[44,45], although other studies have suggested that these trends are unclear among terrestrial vertebrates[32]. As is the case for speciation, then, there is some empirical evidence for a typical direction of the relationship between diversity and extinction—in this case, positive—but the universality of this pattern among vertebrates, too, remains an open question.

Hominin evolution is represented by a well-studied and rich fossil record and occurs across temporal and spatial scales that sit squarely at the expected intersection of climatic and competitive processes[46]. Therefore, exploring how hominin macroevolution fits into the general vertebrate macroevolutionary landscape has the potential to offer new perspectives on longstanding questions in vertebrate evolution, as well as addressing the comparative dearth of explicit research on diversity-dependent macroevolution in the hominin lineage.

Here we ask whether hominins also follow the pattern of diversity-dependent diversification that characterizes many other vertebrate clades. More specifically, we ask: were hominins characterized by negative diversity-dependent speciation and positive diversity-dependent extinction?

At which taxonomic level should these patterns be expected? Negative diversity-dependent speciation and positive diversity-dependent extinction at the level of the hominin clade as a whole would imply either that hominins were characterized by species' inabilities to diverge ecologically from each other, as in Darwinian diversity dependence—and in an extension of Wolpoff's 'culture' argument—or that hominins occupy a bounded set of niches in broader ecological context, as in asymptotic diversity dependence. One possibility is that hominin diversification is not diversity dependent, either because hominin speciation and extinction are purely climate-driven and not determined by diversity-mediated competitive dynamics[2–7], or because the lineage was characterized by consistent ecological divergence, or because a limit to species diversity did not exist or was not reached. A second possibility is that hominins, overall, conform to the expected patterns. This would indicate a powerful and underappreciated role for interspecific competition in hominin evolution.

However, Darwinian diversity dependence[16,37] predicts stronger signals of diversity-dependent dynamics within and not across adaptive grades[15,16,37], as species within adaptive grades should be more ecologically similar to each other. Given that there is strong support for *Homo* having occupied an adaptive grade distinct from earlier hominins[47,48], we contrast the patterns found between *Homo* and Plio-Pleistocene non-*Homo* species (*Australopithecus* and *Paranthropus*). In addition to the two possible patterns described above, this comparison presents a third possibility: conflicting patterns between adaptive grades. Such a pattern will have resonance with the major issue of how far hominin evolution conforms to general evolutionary patterns, and why it might diverge.

**Table 1 | Estimated times of origination and extinction for the models based on the finest-grained occurrence data**

| Species | No preservation prior | | Within-lifetime variability | | Time-based variability | |
|---|---|---|---|---|---|---|
| | Speciation (Ma) | Extinction (Ma) | Speciation (Ma) | Extinction (Ma) | Speciation (Ma) | Extinction (Ma) |
| *Australopithecus afarensis* | 3.70 | 3.00 | 4.18 [4.03, 4.74] | 2.33 [2.22, 2.49] | 3.85 [3.81, 3.90] | 2.35 [2.22, 2.49] |
| *Australopithecus africanus* | 3.00 | 2.40 | 4.33 [3.80, 4.97] | 1.70 [1.80, 2.35] | 3.85 [3.70, 4.22] | 2.07 [1.80, 2.35] |
| *Australopithecus anamensis* | 4.20 | 3.90 | 4.56 [4.29, 4.88] | 3.37 [3.11, 3.55] | 4.44 [4.20, 4.85] | 3.50 [3.30, 360] |
| *Australopithecus bahrelghazali* | 3.58 | 3.58 | 3.32 [3.07, 3.52] | 3.28 [3.04, 3.5] | 3.35 [3.07, 3.57] | 3.24 [2.91, 3.50] |
| *Australopithecus deyiremeda* | 3.50 | 3.30 | 3.60 [3.35, 3.99] | 3.19 [2.80, 3.44] | 3.57 [3.35, 3.99] | 3.20 [2.70, 3.44] |
| *Australopithecus garhi* | 2.50 | 2.45 | 2.52 [2.50, 2.56] | 2.48 [2.44, 2.50] | 2.55 [2.50, 2.70] | 2.45 [2.31, 2.51] |
| *Australopithecus sediba* | 1.98 | 1.98 | 2.08 [1.91, 2.15] | 2.03 [1.89, 2.11] | 2.15 [1.91, 2.43] | 2.00 [1.85, 2.11] |
| *Homo erectus* sensu lato | 1.81 | 0.03 | 2.34 [2.20, 2.48] | 0.01 [0.00, 0.01] | 2.11 [2.02, 2.25] | 0.00 [0.00, 0.01] |
| *Homo floresiensis* | 0.06 | 0.02 | 1.17 [0.80 1.64] | 0.02 [0.01, 0.10] | 1.14 [0.74, 1.83] | 0.04 [0.01, 0.08] |
| *Homo habilis* | 2.35 | 1.65 | 2.75 [2.51, 3.04] | 1.04 [0.76, 1.29] | 2.54 [2.37, 2.85] | 1.23 [1.02, 1.44] |
| *Homo heidelbergensis* | 0.70 | 0.10 | 1.33 [1.00, 1,70] | 0.02 [0.01, 0.05] | 1.32 [0.97, 1,75] | 0.08 [0.02, 0.13] |
| *Homo neanderthalensis* | 0.13 | 0.04 | 0.84 [0.39, 1.31] | 0.01 [0.01, 0.01] | 0.84 [0.39, 1.32] | 0.01 [0.00, 0.02] |
| *Homo rudolfensis* | NA | NA | 2.36 [2.03, 2.73] | 1.31 [0.71, 1.80] | 2.26 [2.03, 2.73] | 1.51 [1.17, 1.82] |
| *Homo sapiens* | 0.20 | 0 | 0.32 [0.29, 0.37] | 0 | 0.33 [0.29, 0.37] | 0 |
| *Paranthropus aethiopicus* | 2.66 | 2.30 | 3.37 [2.92, 3.44] | 1.91 [1.44, 2.24] | 3.21 [2.92, 3.44] | 2.22 [2.03, 2.34] |
| *Paranthropus boisei* | 2.30 | 1.30 | 3.22 [2.64, 3.40] | 0.69 [0.31, 1.13] | 3.14 [2.64, 3.40] | 1.31 [1.15, 1.41] |
| *Paranthropus robustus* | 2.00 | 1.00 | 2.54 [1.81, 2.74] | 0.59 [0.17, 0.92] | 2.22 [1.81, 2.74] | 0.88 [0.70, 0.96] |

Ma, million years ago; NA, not applicable (not present in dataset). Brackets indicate 95% credible interval of dates.

## Results

### Analyses based on speciation and extinction times

To explore whether species diversity predicts species origination and extinction, we ran birth–death models in a validated Bayesian framework[49,50] on five datasets of estimated times of species origination and extinction. The first dataset comprised published first and last appearance dates (FADs and LADs), which are conventionally used as proxies for times of species origination and extinction without accounting for variability in fossil preservation rates. The subsequent four datasets were based on our database of hominin fossil occurrences, recorded at two operational definitions of localities (at the finest-grained occurrence level available (*n* = 385 occurrences) and at the broadest occurrence level (that in which all occurrences at a site complex were merged into a single occurrence; *n* = 267 occurrences)). We applied two sets of explicit fossil preservation rate priors (time-based variability and within-lifetime variability; both models also included between-lineage variability) to these two occurrence datasets, generating four new sets of times of species origination and extinction. As there were no differences in the direction of inferred relationships between these datasets, we report results for both models of preservation from the most fine-grained occurrence level here. Results for the broadest occurrence level are provided in Supplementary Table 1.

Times of speciation and extinction estimated for fine-grained occurrences under (1) lineage- and time-based variability and (2) lineage- and within-species lifetime variability in fossil preservation rates are presented in Table 1. Those for the same analyses across the broadest occurrence level are reported in Supplementary Table 2. Speciation times were significantly different between the new dates estimated under both preservation rate priors and between dates estimated under preservation rate priors and the published FADs and LADs of fossil species (pairwise paired *t*-tests with Bonferroni correction, *P* < 0.05). Extinction times did not differ significantly between dates estimated under both preservation rate priors, but both of these did differ significantly from the published FADs and LADs (pairwise paired *t*-tests with Bonferroni correction, *P* < 0.05). Compared to published FADs, estimates of speciation and extinction times that

account fossil preservation extended species' lifespans. These models estimate that species originated, on average, 0.49 million years earlier (within-lifetime variability) and 0.37 million years earlier (time-based variability) than published dates suggest, and that they went extinct 0.27 million years later (within-lifetime variability) and 0.15 million years later (time-based variability) than published dates suggest (Fig. 1a and Table 1).

**Speciation.** Across each of the three datasets, the results suggest at reasonable confidence that speciation was a negatively diversity-dependent process across the hominin clade as a whole (Fig. 2a and Table 2). The 50% credible intervals did not overlap with 0, although the 95% credible intervals did. In two models, >75% of the posterior distribution was negative. For non-*Homo* species, the signal was more diffuse. Although mean correlation parameter estimates were negative, the 50% credible interval overlapped with 0 in all cases (Fig. 2a). By contrast, the results suggest at reasonable confidence that speciation in *Homo* was positively diversity-dependent across all three datasets (Fig. 2a and Table 2). In the model with no preservation priors, the 50% credible interval overlapped with 0, but the correlation parameter was positive in 67.9% of the iterations. In models incorporating fossil preservation variability, the 50% credible intervals did not overlap with 0, although the 95% credible intervals did (Fig. 2a). In these models, the correlation parameter was positive in 78% and 83.8% of the iterations of the time-based preservation variability model and the within-lifetime preservation variability models, respectively.

**Extinction.** The strongest signal across both processes was that extinction in *Homo* was unexpectedly negatively diversity dependent. For all three models, >75% of the posterior distribution of the diversity correlation parameter was negative, and the 50% credible intervals did not overlap with 0; and in the two models incorporating fossil preservation variability, >99% of the distribution was negative (Fig. 2a and Table 2). The pattern in *Homo* stands in stark contrast to the lack of a strong signal within non-*Homo* or the hominin clade as a whole. Although the mean of the posterior distribution of the correlation

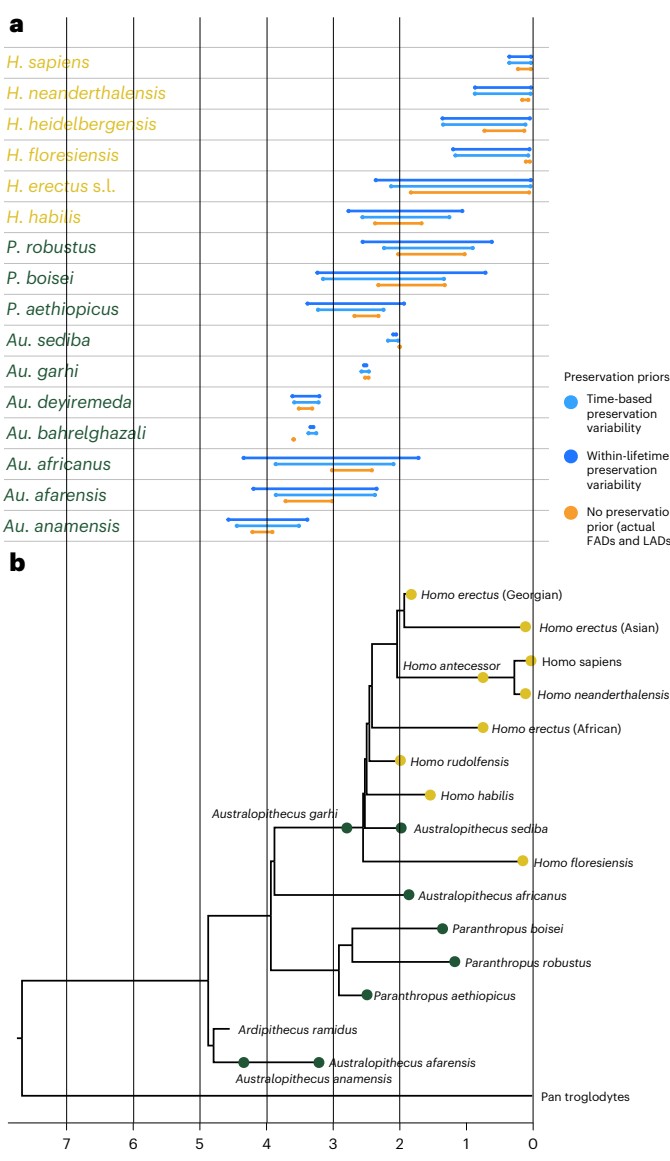

**Fig. 1 | Speciation and extinction dates and phylogeny used in subsequent analyses. a**, Species lifespans, comprising the time between speciation and extinction dates based on three datasets. Orange: published fossil FADs and LADs estimated without taking fossil preservation into account. Light blue: speciation and extinction dates estimated in a Bayesian framework incorporating time-based variability in fossil preservation rates. Dark blue: speciation and extinction dates estimated in a Bayesian framework incorporating within-lineage variability in fossil preservation rates. Note that these taxa are those the published dates and our new database have in common; actual analyses incorporated *Homo ergaster* in the no-preservation-prior dataset and *Homo rudolfensis* in the preservation prior datasets. *Homo erectus* s.l. refers to *Homo erectus* sensu lato. **b**, The Parins-Fukuchi et al.[87] phylogeny used in this study, with species coloured by taxonomic grouping (yellow: *Homo;* green: non-*Homo*).

parameter was positive in all but one model (non-*Homo*, no preservation priors), <75% of the posterior distributions for these models was positive, and the 50% and 95% credible intervals overlapped with 0 in all cases (Fig. 2a and Table 2).

### Phylogeny-based analyses of speciation rate

To explore support for the results from models incorporating variability in fossil preservation versus those from the model without, we developed a complementary but independent analytical approach and calculated the speciation rate across a phylogeny. There was a

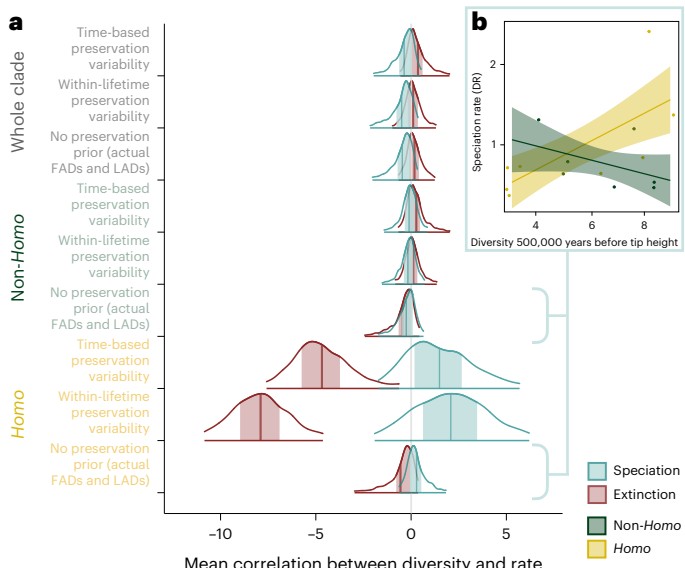

**Fig. 2 | A divergent relationship between macroevolutionary rates and species diversity in *Homo*. a**, Results from PyRate birth–death models, run across three datasets with different fossil preservation priors (no preservation priors: published FADs and LADs; time-based preservation variability, where preservation is allowed to vary every 1 million years−prior applied to fossil occurrence data from three databases; and within-lifetime preservation variability, where preservation rate is allowed to vary across a species' lifespan− prior applied to fossil occurrence data from three databases). In the latter two models, the preservation rate was also allowed to vary between lineages. The posterior distribution of the correlation parameter is shown, with the 50% credible interval shaded and the 95% credible interval indicated by the outline. The mean correlation parameter is indicated by a thick line. Results for speciation are indicated in blue; those for extinction are shown in red. **b**, The relationship between diversity 500,000 years before tip height and speciation rate (tip DR) across the Parins-Fukuchi et al.[87] phylogeny. Shaded area indicates 95% confidence interval.

statistically significant difference in the relationship between diversity and speciation between *Homo* and non-*Homo* groups (phylogenetic generalized least squares (GLS): difference between *Homo* and non-*Homo* regression slope, $P < 0.05$). In non-*Homo* species, speciation rate decreased as a function of diversity, consistent with a negative diversity-dependent speciation regime, whereas *Homo* was characterized by a significant positive relationship between speciation rate and diversity (Fig. 1b). Model outputs are provided in Supplementary Table 3.

The differences between *Homo* and non-*Homo* are unlikely to be the consequence of higher undersampling of non-*Homo* species richness: the same result was obtained across 99% of trees with 12.5% increased non-*Homo* species richness and 98%, 95% and 94% of trees with 25%, 37.5% and 50% increased non-*Homo* species richness, respectively.

The phylogeny-based method performs reasonably well across small datasets; across the small trees simulated under a diversity-dependent process, it correctly inferred negative diversity dependence across 73% of the trees, and this remained 73% when up to 40% species were randomly removed from the phylogenies to simulate incomplete sampling. The method falsely identified a relationship between diversity and speciation across 31% of the simulated constant-rate birth–death trees, and this rose to 41% when up to 40% species were randomly removed from the phylogenies. In the incorrect sample of 31%, negative diversity dependence was inferred across nearly all trees (97%), and this did not change across the phylogenies across which incomplete sampling was simulated. Positive diversity-dependent speciation was incorrectly inferred across 0.02% of the complete simulated

**Table 2 | Relationships between macroevolutionary rate and diversity**

| | | Mean of posterior distribution of correlation parameter | 95% HPD interval | Percentage of posterior distribution greater or lesser than 0 |
|---|---|---|---|---|
| **Speciation** | | | | |
| Whole clade | Time-based preservation variability | −0.40 | −1.67, 0.70 | 73.4% |
| | Within-lifetime preservation variability | −0.50 | −1.97, 0.41 | **81.6%** |
| | No preservation prior | −0.47 | −1.88, 0.37 | **82.0%** |
| Non-*Homo* | Time-based preservation variability | −0.12 | −1.18, 0.99 | 58.4% |
| | Within-lifetime preservation variability | −0.18 | −1.32, 0.83 | 62.9% |
| | No preservation prior | −0.27 | −1.60, 0.69 | 67.2% |
| *Homo* | Time-based preservation variability | 1.47 | −1.85, 5.41 | **78.0%** |
| | Within-lifetime preservation variability | 2.06 | −1.80, 6.24 | **83.8%** |
| | No preservation prior | 0.27 | −0.80, 1.58 | 67.9% |
| **Extinction** | | | | |
| Whole clade | Time-based preservation variability | 0.36 | −0.65, 1.80 | 72.7% |
| | Within-lifetime preservation variability | 0.10 | −0.84, 1.38 | 58.2% |
| | No preservation prior | 0.18 | −0.69, 1.15 | 64.3% |
| Non-*Homo* | Time-based preservation variability | 0.27 | −0.74, 1.72 | 66.3% |
| | Within-lifetime preservation variability | 0.16 | −0.96, 1.16 | 57.6% |
| | No preservation prior | −0.45 | −1.89, 0.61 | **76.4%** |
| *Homo* | Time-based preservation variability | **−4.68** | **−7.76, −0.95** | **99.4%** |
| | Within-lifetime preservation variability | **−7.90** | **−10.82, −4.61** | **100%** |
| | No preservation prior | −0.56 | −2.41, 0.53 | **77.4%** |

The 95% highest posterior density (HPD) interval is shown in bold if it does not overlap with 0. The percentage of the posterior distribution greater or less than 0 (depending on the direction of mean of the posterior distribution) is shown in bold if it is more than 75%. The mean is shown in bold if both the 95% HPD interval does not overlap with 0 and the percentage of the posterior distribution in the direction indicated by the mean is >75%.

constant-rate birth–death trees and 0.3% of the trees across which incomplete sampling was simulated. Taken together, these results suggest that there can be reasonable but not total confidence that non-*Homo* speciation was characterized by negative diversity dependence; there is a 69% chance that it was not a false positive result across a tree generated under a non-diversity-dependent process. There is only a 0.02% chance that the positive diversity-dependent speciation of *Homo* is a methodological artefact.

## Discussion

We investigated whether hominin speciation and extinction are correlated with species diversity, as they are across many—but not all—vertebrate clades. Our results across the clade as a whole suggest that speciation was probably negatively regulated by diversity, and that this was also true for the group comprising *Australopithecus* and *Paranthropus*, although the signal was weaker in this subclade. By contrast, there is reasonably strong evidence that the relationship between speciation and diversity in the genus *Homo* diverges from that in other hominins and many other vertebrates: across two analytical approaches, *Homo* speciation was positively diversity dependent. *Homo* extinction, furthermore, showed a very strongly negatively diversity-dependent pattern, which differs from the lack of a relationship between extinction and diversity found both across hominins as a whole and in the *Australopithecus* and *Paranthropus* subgroup.

From a broader vertebrate perspective, the reasonably strong evidence for negative diversity-dependent speciation across the clade as a whole (mean 79% and 77% of the posterior distributions <0 for all Bayesian models and Bayesian models accounting for fossil preservation bias, respectively; Table 1), paired with the much more diffuse signal of negative diversity dependence in non-*Homo* (mean 62.8% and 60.5% of the posterior distributions <0 for all Bayesian models and

Bayesian models accounting for fossil preservation bias, respectively, and a 69% chance that the patterns across the phylogeny were not false positives) echoes theoretical uncertainty about the taxonomic scale across which diversity dependence should operate[16]. Previous work in higher taxonomic groupings within birds and squamates found evidence of negative diversity-dependent speciation[14,29,31], whereas work within terrestrial vertebrate orders, including Primates, recovered no such relationship[39]. Although our analyses were conducted at a lower scale, we recovered a corresponding pattern, with stronger evidence for diversity-dependent speciation at higher taxonomic scales. Turning to broader questions about whether ecological limits to species diversity are even to be expected[51], the comparison between signal strength at the two scales we report here suggests that limits to diversity should exist at higher taxonomic levels. One potential explanation is that if limits to ecologically similar species diversity exist, subclades may often not reach these individually before extinction begins to outpace speciation. Through a Darwinian diversity-dependent lens, this 'limit' may simply reflect the point at which species are no longer able to diverge ecologically from one another[16] and so has more to do with the evolvability of the clade itself, whereas an asymptotic view holds that there is a set number of niches or limited geographic space to speciate into a priori. We cannot distinguish between these alternatives based on the models we present here, but our results are consistent with clade-wide diversity limits that governed speciation overall but were not quite reached by the non-*Homo* subclade before its extinction.

Notably, neither non-*Homo* subclade extinction nor that across the clade as a whole carried a strong signal of diversity dependence (Table 1). This aligns with previous work reporting an absence of diversity-dependent extinction—for example, the lack of empirical evidence for extinctions in 'saturated' communities following species invasions[52]. Previous work has made a strong case that hominin

extinction is more closely linked to climate than hominin speciation[53], and climate-driven extinction before a theoretical cap on diversity—whether Darwinian or asymptotic—was reached can explain these results.

Contrary to expectations, we found reasonably strong evidence for positively diversity-dependent speciation in *Homo* (mean 76.6% and 80.9% of the posterior distributions >0 for all Bayesian models and Bayesian models accounting for fossil preservation bias, respectively, and a 0.02% chance that the patterns across the phylogeny were false positives). This pattern is much rarer than negatively diversity-dependent speciation and non-diversity-related speciation across all forms of life, having been reported for only a few groups, including island-dwelling beetles[33], plants and arthropods[34], with this last case being contentious[35,36]. In the vertebrate context, then, our results not only expand the set of reported associations between diversity and speciation but, crucially, underscore previous findings that the relationship between speciation and diversity is complex[39,52,54].

This complexity is no doubt at least in part the consequence of the fact that both extrinsic factors and intrinsic traits modify any feedback loop between species diversity and speciation[18,55]. One explanation that incorporates both is repeated dispersals. *Homo* is the only hominin genus to expand its range outside of Africa[56,57], and recurrent expansions into new habitats that promote new adaptations, and therefore speciation, while source populations persist, will result in a positive correlation between speciation rate and diversity. This correlation may reflect no causal relationship between competition and speciation at all, but a complementary model is that high levels of competition between closely related and ecologically similar taxa in the 'source' location drove dispersal in the first place[58,59]. For example, Carbonell and colleagues[60] suggested that the earliest European hominins were competitively displaced from Africa by populations that developed the Acheulian.

A second and non-mutually exclusive explanation is that 'diversity begets diversity'[32]—that is, that existing species provide evolutionary pressure and opportunities for (the evolution of) new species[61–63]. Increased species diversity may produce interactions that can promote speciation and can extend species' lifespans providing that the activities of the species upon which the niche of another is built persist. Species may create novel ecological opportunities for new species to exploit through ecosystem modification[64–66]. Ecosystem modification is particularly likely to lead to evolutionary consequences if it increases structural and resource heterogeneity[64]. Although the time-averaged nature of the fossil record makes it difficult to reconstruct the ecological effects of hominin behaviour, it is not unlikely that hominins, particularly those belonging to the genus *Homo*, were ecosystem engineers[67,68], and that the ecological opportunities afforded by their behaviour promoted the appearance of novel hominin species. Behaviours that may have contributed particularly strongly to such dynamics are the use of fire[69], which can cause widespread landscape modification, and the adoption of active and intensive hunting, which will have exerted new pressures on the distribution and population sizes of hominin prey[68].

An adaptive 'trait' potentially critical to repeated geographic expansions—whether caused by competition or not—as well as creating ecological opportunities for new species and exploiting ecological opportunities afforded by other hominins, is technology[70,71]. Of course, lithic technology predates *Homo*[72], but what can be said with some certainty is that neither *Australopithecus* and *Paranthropus*, nor other vertebrates, rely on stone tool technology to the same degree as *Homo*[73]. The suggestion that repeated and ratcheted technological innovation promoted speciation stands in stark contrast to Wolpoff's single species hypothesis[1].

However, it may be that *Homo* speciation merely appears to be positively diversity dependent because non-*Homo* taxa were present at the time of *Homo* species origination, and that it is not necessarily the presence of these taxa that resulted in *Homo* speciation. What is incongruous about this explanation, in the context of the evidence for negative diversity dependent speciation across the clade as a whole, is why the presence of these other species did not restrict *Homo* speciation as predicted. This overlap hints at competitive displacement from occupied niches, rather than the opportunistic replacement documented in, for example, carnivores[74].

*Homo* is characterized, finally, by negative diversity-dependent extinction (mean 92.3% and 99.7% of the posterior distributions <0 for all Bayesian models and Bayesian models accounting for fossil preservation bias, respectively). To the best of our knowledge, this pattern has not been documented in other clades. It may be the case that negative diversity-dependent extinction is a statistical artefact of coeval extinction events unrelated to diversity—presumably precipitated by climate change[53]. Although we cannot rule this out, because we did not explicitly contrast the effects of climate and diversity in our models, there are problems with a climate-only explanation for this pattern. Given that *Homo* evolution occurred during a period of increased climatic deterioration and change[5,75,76], and that estimated extinction times did not occur at exactly the same time, and both before and after episodes of major cooling (Table 1), it is difficult to pinpoint a single climatic event that underlies *Homo* extinctions. We propose two alternative explanations for this pattern. First, it may be the case that repeated innovations in adaptive strategies in *Homo* resulted in coeval competitive replacement of a number of species by a single innovator species. For instance, shifts in life history and dietary strategy within *Homo erectus* sensu lato[77] may have allowed this taxon to replace competitively early forms of *Homo*, and the same may have been true for the cognitive and behavioural innovations of *Homo sapiens* relative to Late Pleistocene hominins[78]. This echoes a recent point made by Bokma and colleagues[79]. A related explanation for late Pleistocene replacement by *Homo sapiens* is that most roughly coeval hominin extinctions are caused by climate, but that the extreme generalism of *Homo sapiens* prevents late-surviving forms from speciating[16]. These mechanisms, which may interact with climatic and environmental shifts which stimulated the adaptive novelties, would produce a negative diversity signal.

Accounting for variability in fossil preservation rates within and between lineages, and across time, resulted in longer estimated species lifespans (Fig. 1a) than those based on published FADs and LADs, and these extended lifespans are more in line with the mammalian average of 1 million years[80,81]. That actual hominin fossil FADs and LADs do not fully represent species' lifespans aligns with findings that hominin fossils are comparatively rare within mammal assemblages[82]. Further implications of these results relate to phylogeny: there are three sets of conventionally hypothesized ancestor–descendant relationships that our new origination and extinction times suggest cannot have occurred unless in the context of non-Hennigian speciation[83], in which the ancestral species persists alongside its daughter species. *Paranthropus boisei* originated ~0.1 million years after its putative ancestor *Paranthropus aethiopicus*[84], and they overlapped temporally for >0.5 million years. This latter pattern is also true for *Homo heidelbergensis* and *Homo neanderthalensis*, and for *Australopithecus anamensis* and *Australopithecus afarensis*[85]. In the case of *Au. anamensis* and *Au. afarensis*, our results echo recent evidence for the contemporaneity of the two species[86]. Further, the earlier origination date of *Homo floresiensis* aligns with that inferred across the Parins-Fukuchi et al.[87] phylogeny. Overall, these new datasets underscore previous calls[82] to account for incomplete sampling in analyses of hominin macroevolution. Our results suggest that using conventional FADs and LADs underestimates species' temporal ranges, with attendant problems for the validity of conclusions.

Evolution is clearly a pluralistic process, with the attendant expectation that climatic processes and competition both influence vertebrate macroevolution. Disentangling their relative roles is an area of ongoing research[13,88,89], and the first step in doing so in

hominins was addressing the comparative dearth of explicit research on diversity-dependent macroevolution. The evidence for negative diversity dependence in speciation across the clade overall and the strong and unexpected evidence for the part that interspecific competition may have played in both speciation and extinction in the genus *Homo* are difficult to reconcile with conventional models that place exclusive emphasis on the role of climate in hominin macroevolution. Ultimately, the climatic 'Court Jester' must set the stage upon which the 'Red Queen' of interspecific competition 'dances'[90], and our results point to a need to further explore the relationship between climate and competition, and how this relationship drove macroevolution, in our own lineage. Finally, an important effect of the inferred longer species lifespans is notably extended periods of temporal overlap between sympatric species, such as *P. boisei* and early *Homo* in East Africa[19] as well as *Paranthropus robustus* and *Australopithecus africanus* in South Africa. Extended periods of sympatry provide the context for interspecific competition at smaller scales, with effects at equivalent scales, such as microevolutionary morphological evolution driven by competition-mediated niche separation[10,13,91]. Our results, then, point to the need to further explore the possible effects of interspecific competition at all scales[16].

## Conclusion

In vertebrates, speciation is often negatively diversity dependent, and extinction is expected to show positive diversity dependence. Our results are consistent with diversity limits at the level of the hominin tribe that governed speciation in hominins overall, and with these not being quite reached by the *Australopithecus* and *Paranthropus* subclade before its extinction. There was no signal of diversity dependence in *Australopithecus* and *Paranthropus* subclade extinction, or that of hominins overall: this is concordant with climate playing a bigger part in hominin extinction than speciation. *Homo* emerged as an evolutionary outlier amongst its hominin and vertebrate relatives. There is strong evidence that *Homo* extinction was negatively diversity dependent. Whether this reflects a process of repeated replacements of numerous older forms by more modern species of *Homo* or simply a correlation with pulsed climatic events, or a more complex relationship between climate and diversity, are new questions raised by these results. Finally, speciation in *Homo* was found to be relatively robustly positively diversity dependent across two analytical approaches. We argue that the comparatively unusual pattern of positive diversity-dependent speciation we report is concordant with a set of underappreciated and non-mutually exclusive drivers of speciation in *Homo*: interspecific competition, repeated geographic expansions potentially driven by interspecific competition, and ecosystem engineering by other members of *Homo* opening up new niches. Whatever the exact processes driving these patterns, the results presented here suggest that *Homo* was characterized by comparatively unusual and unexpected macroevolutionary dynamics.

## Methods
### Data collection
**Occurrence data and FADs and LADs.** Fossil occurrence data were obtained in November 2023 from the Paleobiology Database (https://paleobiodb.org/#/), using a taxon search for 'hominin'; the NOW Database[92]; and the ROCEEH ROAD Database[93]. The taxonomy of the Paleobiology Database occurrence data was checked for spelling errors using the PyRate 'check_names' function[49] and manually for synonyms. Occurrence data from the three databases were merged. Duplicates were identified manually, and records with most up-to-date age estimates were retained. If occurrences did not have a specified accession number, duplicates were identified based on location (geological formation and/or member, latitude and longitude) in combination with inspection of specified source publications (if available). To account for differences in the three databases' approach to defining occurrence

localities (for example, *Au. afarensis* at Laetoli comprises two entries in the Paleobiology Database, both of which are composites of more than two find spots, whereas all find spots are separate entries in the NOW database), we took a hierarchical approach to recording occurrences, recording 'Site complex' (for example, the Woranso-Mille palaeoanthropological research area), 'Site' (for example, Taung), 'Subsite' (for example, localities or surface find spots within a 'site'; subsite 'type' was also recorded), 'Formation' (for example, Koobi Fora) and 'Stratigraphic unit' (for example, Member 4). Not all occurrences had information for all variables: for example, the Mauer site is not part of a larger 'Site' complex. We supplemented and updated the merged database with occurrence information obtained from literature reviews of papers published after 2016 and cross-checked our database with occurrence information supplied in published overviews of research where available[84,94,95].

Species' published FADs and LADs, which are conventionally taken as speciation and extinction 'times', were taken from Wood and Boyle[96] and supplemented with dates of more recently published species in the manner described by van Holstein and Foley[97].

**Phylogeny.** We used the phylogeny with the best Akaike information criterion score from Parins-Fukuchi et al.[87]. In contrast to other hominin phylogenies[98–100], this phylogeny combines probabilistic models of morphological evolution and fossil preservation to recover anagenetic relationships between hominin species. It therefore uniquely recovers ancestor–descendant relationships that are (1) likely to be more realistic than those on phylogenies that do not incorporate them, and (2) broadly accepted based on morphological evidence alone (for example, between *Au. anamensis* and *Au. afarensis*[85]).

The species diversification rate, tip DR, of Jetz et al.[101], which calculates the tip-specific speciation rate, was calculated for every tip using R code from Upham et al.[102]:

$$DR = \left( \sum_{j=1}^{N_i} l_j \frac{1}{2^{j-1}} \right)^{-1} \tag{1}$$

where DR is the tip DR for species *i*, $N_i$ is the number of edges between species *I* and the root, and *l* is the length of edge *j* (with *j* = 1 being the edge closest to the extant tip). For each tip, the number of extant species at 500,000 years before the tip height was obtained using the 'getExtant' function in the phytools package[103].

### Analyses
**Analyses based on speciation and extinction times.** To determine whether speciation and extinction times were correlated with species diversity, we ran birth–death models, with diversity as predictor, in a validated Bayesian framework[49,50] on five datasets with estimated times of species origination and extinction. The first dataset was based on the conservative FADs and LADs estimated by Wood and Boyle[96], with additions from van Holstein and Foley[97], and thus incorporates no variability in fossil preservation rates. The subsequent four datasets were based on our new hominin occurrence database. From these data, we estimated four new sets of times of speciation and extinction with two sets of explicit fossil preservation rate priors and two operational definitions of localities (at the finest-grained occurrence level available (*n* = 385 occurrences) and at the broadest occurrence level (that is, with all occurrences at a site complex merged into a single occurrence; *n* = 267 occurrences). As there were no differences in the direction of inferred relationships between these datasets, we report results for both models of preservation from the most fine-grained occurrence level; results for the broadest occurrence level are provided Supplementary Table 1.

In the first dataset, we modelled fossil preservation as a function of a time-variable Poisson process. Preservation rates were allowed

to vary every 1 million years. In the second dataset, we allowed fossil preservation to vary over the course of a species' lifespan by modelling it with a non-homogeneous Poisson process of preservation[49]. This allowed us to take into account that fossils are less likely to form at the start and end of a species' lifespan, as the number of individuals belonging to a species is low. In both datasets, fossil preservation was also allowed to vary between lineages by incorporating a gamma model of rate heterogeneity[49]. We generated ten replicates of estimated times of species origination and extinction for both preservation regimes using the Reversible Jump MCMC algorithm in the python programme PyRate[49], to incorporate dating uncertainty into the results[104]. All analyses described below were then performed on the ten replicates, and results were joined into a single posterior sample.

We generated lineage-through-time estimates for all three datasets in PyRate. We then applied a PyRate birth–death model in which *Homo* and non-*Homo* speciation and extinction rates were determined by an exponential correlation to a time-variable predictor, in this case clade-wide lineage-through-time estimates for each set of times of origination and extinction. To compare these results with the pattern across the whole clade, we ran an exponential diversity-dependent birth–death model, in which the whole clade's own diversity was used as the predictor variable. As the sample size was inevitably relatively small, we ran each model for 2,000,000 iterations, sampling every 1,000 iterations.

**Analyses based on phylogeny.** These analyses were performed in R 4.01 (ref. [105]). To explore further the relationship between speciation and diversity—and, in particular, the difference between *Homo* and non-*Homo* species—we ran phylogenetic GLS regressions to determine whether there were differences between *Homo* and non-*Homo* in the relationship between speciation rates and previous clade-wide diversity:

$$DR = SD \times \text{taxonomic group} \qquad (2)$$

where DR is the tip DR and SD is the phylogeny-based species diversity at 500,000 years before the tip.

The phylogenetic correlation structure of residual error in the phylogenetic GLS was accounted for in the nlme 'correlation' argument[106]. The model assumed a Brownian motion model for residual error structure, following previous work on regressions including speciation rates[101,107,108]. Non-contemporaneity of tips was represented in the nlme argument 'weights'.

To test the ability of the phylogeny-based approach described above to correctly distinguish between diversity-dependent and non-diversity-dependent speciation, we simulated 1,000 phylogenies under a constant-rate birth–death process using the 'pbtree' function in the phytools package[103], preserving extinct tips, and repeated the analyses described above for equation (2) to estimate how often diversity-dependent speciation was erroneously inferred across non-ultrametric trees generated under a non-diversity-dependent process. We then generated 1,000 phylogenies simulated under a diversity-dependent regime using the 'ddsim' function in the DDD package[109]. These phylogenies were simulated with birth and death rates and carrying capacities randomly drawn from a normal distribution with means that produced trees with similar tip numbers to those of the Parins-Fukuchi et al.[87] phylogeny in a trial run, and with a maximum tree height of 7, so as to produce similarly small phylogenies to the Parins-Fukuchi et al.[87] tree. We then randomly removed up to 40% of tips and repeated the analyses to investigate the sensitivity of results to incomplete sampling.

We also tested the sensitivity of the results of phylogeny-based analyses using equation (2) to the increased probability of species discovery towards the present[82], which could have resulted in underrepresentation of non-*Homo* species relative to the younger *Homo* species included in the analyses. To do so, we generated 4,000 phylogenies with up to +50% non-*Homo* species added in random locations and with random tip heights to the original Parins-Fukuchi et al.[87] phylogeny, using the 'bind_tip' function in the phytools package[103]. We repeated the analyses described above for equation (2) and calculated the proportion of trees across which the original results were maintained.

## Reporting summary

Further information on research design is available in the Nature Portfolio Reporting Summary linked to this article.

## Data availability

All data are available on figshare: https://figshare.com/s/46fe37e09047513e31b0 (ref. [110]).

## Code availability

All code is available on figshare: https://figshare.com/s/46fe37e09047513e31b0 (ref. [110]).

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

## Acknowledgements

We thank J. Saers and M. Mirazón-Lahr for insightful discussion of this manuscript. This work was supported by a Clare College (University of Cambridge) Junior Research Fellowship (2022-2025) awarded to L.v.H.

## Author contributions

L.v.H. conceived the study, collected data, performed analyses, interpreted results and wrote the manuscript. R.A.F. contributed to study conceptualization and interpretation of results and critically edited the manuscript.

## Competing interests

The authors declare no competing interests.

## Additional information

**Correspondence and requests for materials** should be addressed to Laura A. van Holstein.

# Reporting Summary

## Statistics

For all statistical analyses, confirm that the following items are present in the figure legend, table legend, main text, or Methods section.

| n/a | Confirmed | |
|---|---|---|
| ☐ | ☒ | The exact sample size (*n*) for each experimental group/condition, given as a discrete number and unit of measurement |
| ☒ | ☐ | A statement on whether measurements were taken from distinct samples or whether the same sample was measured repeatedly |
| ☐ | ☒ | The statistical test(s) used AND whether they are one- or two-sided<br>*Only common tests should be described solely by name; describe more complex techniques in the Methods section.* |
| ☐ | ☒ | A description of all covariates tested |
| ☐ | ☒ | A description of any assumptions or corrections, such as tests of normality and adjustment for multiple comparisons |
| ☐ | ☒ | A full description of the statistical parameters including central tendency (e.g. means) or other basic estimates (e.g. regression coefficient) AND variation (e.g. standard deviation) or associated estimates of uncertainty (e.g. confidence intervals) |
| ☐ | ☒ | For null hypothesis testing, the test statistic (e.g. *F*, *t*, *r*) with confidence intervals, effect sizes, degrees of freedom and *P* value noted<br>*Give P values as exact values whenever suitable.* |
| ☐ | ☒ | For Bayesian analysis, information on the choice of priors and Markov chain Monte Carlo settings |
| ☒ | ☐ | For hierarchical and complex designs, identification of the appropriate level for tests and full reporting of outcomes |
| ☒ | ☐ | Estimates of effect sizes (e.g. Cohen's *d*, Pearson's *r*), indicating how they were calculated |

*Our web collection on statistics for biologists contains articles on many of the points above.*

## Software and code

Policy information about availability of computer code

| Data collection | a) Occurrence data and first- and last appearance dates |
|---|---|
| | Fossil occurrence data were obtained from (1) the Paleobiology Database, using the taxon search for 'hominin', (2) the NOW Database, and (3) the ROCEEH ROAD Database, in November 2023. Taxonomy of the Paleobiology Database occurrence data was checked for spelling errors using the PyRate "check_names" function52, and manually for synonyms. Occurrence data from the three databases were merged. Duplicates were identified manually, and record with most up-to-date age estimates retained. If occurrences did not have a specified accession number, duplicates were identified based on location (geological formation and/or member, latitude and longitude) in combination with inspecting specified source publications (if available). To account for differences in the three databases' approach to defining occurrence localities (e.g., Australopithecus afarensis at Laetoli comprises two entries in the Paleobiology Database, both of which are composites of >2 find spots, whilst all find spots are separate entries in the NOW database), we took a hierarchical approach to recording occurrences, recording 'Site complex' (e.g., the Woranso-Mille paleoanthropological research area), 'Site' (e.g., Taung), 'Subsite' (e.g., localities or surface find spots within a 'site'; subsite 'type' was also recorded), 'Formation' (e.g., Koobi Fora), 'Stratigraphic unit' (e.g., Member 4). Not all occurrences have information for all variables: for example, the Mauer site is not part of a larger 'Site complex'. We supplemented and updated the merged database with occurrence information obtained from literature reviews of papers published after 2016, and cross-checked our database with occurrence information supplied in published overviews of research where available. |
| | Species' published first appearance dates (FADs) and last appearance dates (LADs), which are conventionally taken as speciation and extinction 'times', were taken from Wood & Boyle and supplemented with dates of more recently published species in the manner described in van Holstein & Foley. |
| | b) Phylogeny |
| | We used the phylogeny with the best Akaike information criterion score from Parins-Fukuchi et al. In contrast to other hominin phylogenies, |

this phylogeny combines probabilical models of morphological evolution and fossil preservation to recover anagenetic relationships between hominin species. It therefore uniquely recovers ancestor-descendant relationships that are (i) likely more realistic than those on phylogenies that do not incorporate them, and (ii) broadly accepted based on morphological evidence alone (for example, between Australopithecus anamensis and Australopithecus afarensis62).

c) Data from phylogeny: speciation rate and previous diversity for each tip
Jetz et al.'s tip DR, which calculates tip-specific speciation rate, was calculated for every tip.
For each tip, the number of extant species at 500k years before the tip was obtained using the "getExtant" function in the phytools package.

**Data analysis**

a) Analyses based on speciation and extinction times: Do speciation and extinction times correlate with species diversity?
We ran birth-death models, with diversity as predictor, in a validated Bayesian framework on five datasets with estimated times of species origination and extinction. The first dataset was based on the conservative FADs and LADs estimated by Wood and Boyle, with additions from van Holstein and Foley and thus incorporates no variability in fossil preservation rates. The subsequent four datasets were based on our new hominin occurrence database. From these data, we created four new sets of estimated times of speciation and extinction with two sets of explicit fossil preservation rate priors and two operational definitions of localities (at the finest-grained occurrence level available [n=385 occurrences], and at the broadest occurrence level [i.e., in which all occurrences at a site complex were merged into a single occurrence; n=267 occurrences]). As there are no differences in the direction of inferred relationships between these datasets, we report results for both models of preservation from the most fine-grained occurrence level, and results for broadest occurrence level are provided Supplementary Table 1.

In the first dataset, we modelled fossil preservation as a function of a time-variable Poisson process. Preservation rates were allowed to vary every 1 million years. In the second dataset, we allowed fossil preservation to vary over the course of a species' lifespan by modelling it with a homogeneous Poisson process of preservation (NHPP). This allowed us to take into account that fossils are less likely to form at the start and end of a species' lifespan, as the number of individuals belonging to a species is low. In both datasets, fossil preservation was also allowed to vary between lineages by incorporating a Gamma model of rate heterogeneity. We generated ten replicates of estimated times of species origination and extinction for both preservation regimes using the Reversible Jump MCMC algorithm in the python programme PyRate to incorporate dating uncertainty into the results. All analyses described below were then performed on the 10 replicates, and results were joined into a single posterior sample.

We generated lineage-through-time estimates for all three datasets in Pyrate. We then applied the PyRate exponential birth-death model, with clade-wide lineage-through-time estimates as the predictor, to the estimated speciation and extinction times of Homo and non-Homo species. To compare these results with the pattern across the whole clade, we ran an exponential diversity-dependent birth-death model, in which the whole clade's own diversity is used as the predictor variable. Because the sample size is inevitably relatively small, we ran each model for 2,000,000 iterations, sampling every 1,000 iterations.
b) Analyses based on phylogeny: Is variation in speciation rate predicted by species diversity?
These analyses were performed in R 4.0167. To explore the relationship between speciation and diversity—and in particular, the difference between Homo and non-Homo species further, we ran phylogenetic generalized least squares (GLS) regressions to ask whether there are differences between Homo and non-Homo in the relationship between speciation rates and previous clade-wide diversity. The phylogenetic correlation structure of residual error in the phylogenetic GLS was accounted for in the nlme "correlation" argument. The model assumed a Brownian motion model for residual error structure, following previous work on regressions including speciation rates . Non-contemporaneity of tips was represented in the nlme argument "weights".

To test the ability of the phylogeny-based approach described above to correctly distinguish between diversity-dependent and non-diversity dependent speciation, we simulated 1000 phylogenies under a constant-rate birth-death process using the "pbtree" function in the phytools package64, preserving extinct tips, and repeated the analyses described above for equation (1) to estimate how often diversity-dependent speciation is erroneously inferred across non-ultrametric trees generated under a non-diversity-dependent process. We then generated 1000 phylogenies simulated under a diversity-dependent regime using the "ddsim" function in the DDD package. These phylogenies were simulated with birth and death rates and carrying capacities randomly drawn from a normal distribution with means that produced trees with similar tip numbers to the Parins-Fukuchi et al.58 phylogeny in a trial run, and with a maximum tree height of 7, so as to produce similarly small phylogenies to the Parins-Fukuchi et al.58 tree. We then randomly removed up to 40% of tips and repeated the analyses to investigate the sensitivity of results to incomplete sampling.

We also tested the sensitivity of the results of phylogeny-based analyses using equation (2) to the increased probability of species discovery towards the present, which could have resulted in the underrepresentation of non-Homo species relative to the younger Homo species included in the analyses. To do so, we generated 4000 phylogenies with up to +50% non-Homo species added in random locations and with random tip heights to the original Parins-Fukuchi et al.58 phylogeny using the "bind_tip" function in the phytools package. We repeated the analyses described above for equation (2) and calculated the proportion of trees across which the original results were maintained.

For manuscripts utilizing custom algorithms or software that are central to the research but not yet described in published literature, software must be made available to editors and reviewers. We strongly encourage code deposition in a community repository (e.g. GitHub). See the Nature Portfolio guidelines for submitting code & software for further information.

# Data

Policy information about availability of data

All manuscripts must include a data availability statement. This statement should provide the following information, where applicable:
- Accession codes, unique identifiers, or web links for publicly available datasets
- A description of any restrictions on data availability
- For clinical datasets or third party data, please ensure that the statement adheres to our policy

All data and code will be made available on FigShare upon publication.

## Research involving human participants, their data, or biological material

Policy information about studies with human participants or human data. See also policy information about sex, gender (identity/presentation), and sexual orientation and race, ethnicity and racism.

| | |
|---|---|
| Reporting on sex and gender | N/A |
| Reporting on race, ethnicity, or other socially relevant groupings | N/A |
| Population characteristics | N/A |
| Recruitment | N/A |
| Ethics oversight | N/A |

Note that full information on the approval of the study protocol must also be provided in the manuscript.

# Field-specific reporting

Please select the one below that is the best fit for your research. If you are not sure, read the appropriate sections before making your selection.

☐ Life sciences      ☐ Behavioural & social sciences      ☒ Ecological, evolutionary & environmental sciences

For a reference copy of the document with all sections, see nature.com/documents/nr-reporting-summary-flat.pdf

# Ecological, evolutionary & environmental sciences study design

All studies must disclose on these points even when the disclosure is negative.

| | |
|---|---|
| Study description | We calculated speciation rates using two approaches -- across a phylogeny, and from fossil occurrence data in a Bayesian framework -- and then asked whether speciation rates correlate with diversity. We did the same for extinction within the Bayesian framework. |
| Research sample | All hominin species |
| Sampling strategy | N/A |
| Data collection | All data were taken from previous publications (i.e., fossil occurrence data, first appearance date data, phylogeny). |
| Timing and spatial scale | N/A |
| Data exclusions | No data were excluded from the study. |
| Reproducibility | N/A |
| Randomization | N/A |
| Blinding | N/A |

Did the study involve field work?    ☐ Yes    ☒ No

# Reporting for specific materials, systems and methods

We require information from authors about some types of materials, experimental systems and methods used in many studies. Here, indicate whether each material, system or method listed is relevant to your study. If you are not sure if a list item applies to your research, read the appropriate section before selecting a response.

## Materials & experimental systems

| n/a | Involved in the study |
|---|---|
| ☒ | Antibodies |
| ☒ | Eukaryotic cell lines |
| ☐ | ☒ Palaeontology and archaeology |
| ☒ | Animals and other organisms |
| ☒ | Clinical data |
| ☒ | Dual use research of concern |
| ☒ | Plants |

## Methods

| n/a | Involved in the study |
|---|---|
| ☒ | ChIP-seq |
| ☒ | Flow cytometry |
| ☒ | MRI-based neuroimaging |

## Palaeontology and Archaeology

| | |
|---|---|
| Specimen provenance | N/A - all data comes from previous publications |
| Specimen deposition | N/A |
| Dating methods | N/A |

☐ Tick this box to confirm that the raw and calibrated dates are available in the paper or in Supplementary Information.

| | |
|---|---|
| Ethics oversight | N/A |

Note that full information on the approval of the study protocol must also be provided in the manuscript.

