## [Peer Review File · Nature Ecology & Evolution]

Peer Review Information

Journal: Nature Ecology & Evolution

Manuscript Title: Rare patterns of diversity-dependent speciation and extinction in the genus Homo

Corresponding author name(s): Laura A. van Holstein

Editorial Notes:

Reviewer Comments & Decisions:

Decision Letter, initial version:

22nd September 2022

Dear Laura

Your Article, "Speciation patterns in Homo diverge from those of other vertebrates and hominins: implications for interspecific competition's role in macroevolution" has now been seen by three reviewers. The reports are what I would term cautiously enthusiastic--by which I mean that although they identify quite substantial concerns that need to be addressed (such as accounting for sampling bias, robustness checks, and some re-analysis), they are enthusiastic about the questions you are asking and find the attempt to answer them worthwhile. In light of these comments, we cannot accept the manuscript for publication, but would be very interested in considering a revised version that addresses these serious concerns.

We hope you will find the reviewers' comments useful as you decide how to proceed. If you wish to submit a substantially revised manuscript, please bear in mind that we will be reluctant to approach the reviewers again in the absence of major revisions.

If you choose to revise your manuscript taking into account all reviewer and editor comments, please highlight all changes in the manuscript text file.

* Include a "Response to reviewers" document detailing, point-by-point, how you addressed each referee comment. If no action was taken to address a point, you must provide a compelling argument. This response will be sent back to the referees along with the revised manuscript.

* If you have not done so already we suggest that you begin to revise your manuscript so that it conforms to our Article format instructions at <http://www.nature.com/natecolevol/info/final-submission>. Refer also to any guidelines provided in this letter.

2* Include a revised version of any required reporting checklist. It will be available to referees (and, potentially, statisticians) to aid in their evaluation if the manuscript goes back for peer review. A revised checklist is essential for re-review of the paper.

[REDACTED]

If you wish to submit a suitably revised manuscript we would hope to receive it within 6 months. If you cannot send it within this time, please let us know. We will be happy to consider your revision so long as nothing similar has been accepted for publication at Nature Ecology & Evolution or published elsewhere.

Nature Ecology & Evolution is committed to improving transparency in authorship. As part of our efforts in this direction, we are now requesting that all authors identified as 'corresponding author' on published papers create and link their Open Researcher and Contributor Identifier (ORCID) with their account on the Manuscript Tracking System (MTS), prior to acceptance. This applies to primary research papers only. ORCID helps the scientific community achieve unambiguous attribution of all scholarly contributions. You can create and link your ORCID from the home page of the MTS by clicking on 'Modify my Springer Nature account'. For more information please visit please visit www.springernature.com/orcid.

Thank you for the opportunity to review your work.

[REDACTED]

Reviewer expertise:

Reviewer #1: macroevolution, fossil record

Reviewer #2: palaeoanthropology, macroevolution

Reviewer #3: macroevolution, fossil record, computational biology

Reviewers' comments:

2Reviewer #1 (Remarks to the Author):

This ms explores whether hominins have experienced diversity-dependent diversification dynamics which is found in many other (vertebrate) clades. They find evidence for negative diversity-dependence in some clades, but positive diversity-dependence in Homo. While I think the question is interesting and the reported result is surprising, I have concerns about the methodology, and hence doubts about the conclusiveness of the results. I will limit myself to the analysis of speciation rate and its dependence on diversity from phylogenetic information.

My general concern is that the number of data points is limited and experience in this field teaches us that we have very little power for inference. Just looking at Figure 2, I feel that we cannot draw strong conclusions. Related to that I suspect that the negative diversity-dependence may somehow be driven by the fact that these clades become extinct, while the Homo clade has not become extinct. I would like the authors to think about their results in this general way, i.e. if their result is correct, what is it in the data that causes this power to detect a signal?

More specifically, I have concerns about the statistical approach. The authors compute speciation rates using tip DR and then do a GLM. I wonder whether tip DR is a good measure of speciation rate, especially for speciation rate through time. In any case it is still a statistic on which further statistics is done (GLM). I think there is another (arguably better) alternative which calculates the likelihood of the tree, under a model of diversity-dependent speciation (and extinction). This is quite simple for a complete tree, i.e. a tree with all extinct branches in it. I believe that the methods the authors use now also seem to make the assumption that we have complete information. Suppose we have a tree with branching or extinction events at subsequent times t_i , where $i = 1, 2, 3, \dots$ and t_0 is the stem age of the tree. We also introduce $\delta(t_i)$ as follows: if event i is a speciation event then $\delta(t_i) = 1$, otherwise $\delta(t_i) = 0$. Let us assume that the speciation rate and extinction rate depend on the number of species at time t_i through some functions $\lambda(n_i)$ and $\mu(n_i)$, respectively, where n_i is the number of species at t_i . Without any extinctions we have $n_i = i$ if we follow the phylogeny, but one can also use information on the number of species at any point in time. If there is no change in number of species between any two events in the tree, the likelihood is then given by:

$$L = \prod_{i=1}^N e^{-n_i(\lambda(n_i) + \mu(n_i))(t_i - t_{i-1})} (\lambda(n_i))^{\delta(t_i)} (\mu(n_i))^{1 - \delta(t_i)}$$

[Latex for this is:
$$L = \prod_{i=1}^N e^{-n_i(\lambda(n_i) + \mu(n_i))(t_i - t_{i-1})} (\lambda(n_i))^{\delta(t_i)} (\mu(n_i))^{1 - \delta(t_i)}$$
]

One can now use Maximum Likelihood methods or Bayesian statistics to estimate the parameters that govern the functions $\lambda(n)$ and $\mu(n)$. For instance, one can set $\lambda(n) = \max(0, \lambda_0(1 - n/K))$ and $\mu(n) = \mu_0$ as is typically done in diversity-dependent models, and parameters λ_0 , μ_0 and K are estimated. But to allow for both negative and positive diversity-dependence, one could simply take $\lambda(n) = \max(0, \lambda_0 + a n)$ for some parameter a which can be positive or negative. If not all extinctions are observed, this can be accounted for by considering the likelihood of the fossilized birth-death model.

I would like the authors to consider this approach which is quite standard in the literature.

Reviewer #2 (Remarks to the Author):

Key results: This has potential to be a very important contribution because it explores a problem that has received far too little attention. A massive body of literature has examined the role of climate-driven environmental change in shaping macroevolutionary trends within fossil hominins (Court Jester hypotheses). The present paper diverges from this literature by shifting the focus to biotic interactions (Red Queen hypothesis), specifically intraspecific competition, which is typically invoked to explain density-dependent speciation patterns various taxonomic groups. This paper presents an important and long overdue counterpoint to the numerous Court Jester papers in paleoanthropology, and I could easily see a version of it becoming a key citation within the vast literature about the drivers of macroevolutionary change within the hominin clade. It is likely to open the door to new avenues of research. Likewise, the observation that *Homo* shows a unique pattern (speciation rates increase with diversity) compared to other hominin genera is going to capture people's attention.

Validity: As much as I appreciate what the authors set out to accomplish, I have major concerns about how the incomplete and unevenly sampled fossil record has influenced several analyses that are central to the paper. Dealing with this will require that the authors rethink their analytical approach, which could be quite challenging...or fun depending on how you look at it! In its current form, which does not attempt to control for sampling problems, I'm not convinced that many of the results are telling us about macroevolutionary dynamics.

(1) The authors begin by asking whether hominin first appearance dates (FADs) are predicted by hominin diversity in preceding time bins, the expectation being that periods of lower diversity should be followed by a greater number of FADs. This is tricky because sampling effort has a strong effect on hominin diversity (Maxwell et al., 2018) and because incomplete sampling means observed FADs are too young (Faith et al., 2021). This is true for any fossil taxon, but especially hominins because of their rarity in the fossil record. The authors demonstrate some awareness of the problem (line 91-93) but proceed with the analysis without doing anything about it. It is important that the authors acknowledge the complications inherent to the data they are working with by drawing from methodological approaches that circumvent or at least ameliorate sampling problems associated with the fossil record.

Note: The expectation that periods of lower diversity should be followed by a greater number of FADs is also consistent with the effects of sampling. Maxwell et al. (2018) show a general trend of increasing sampling effort in Africa since 7 Ma (though it drops off after 2 Ma). This trend is important since it means that, on average, FADs will tend to be preceded (500 kyr earlier) by a period of lower sampling effort and thus lower richness.

(2) An additional analysis examines whether variation in speciation rates correlate with previous clade-wide and regional diversity. Once again, the problems associated with an incomplete fossil record complicate this analysis. We know that diversity in a given time bin (either clade-wide or at regional scales) is going to be influenced by sampling (Maxwell et al. 2018)—greater sampling effort is

4going to increase the number of known hominin taxa, and it will also expand their observed temporal ranges and thus increase the number of species within a given time bin. I'm less familiar with the DR metric used here (for speciation rates), but my understanding is that DR is highly sensitive to incomplete sampling and missing taxa (see here: [10.1002/ajb2.1479](https://doi.org/10.1002/ajb2.1479)). Thus, I'm not sure whether correlations between speciation rates and diversity are telling us about macroevolutionary dynamics since both variables are very sensitive to sampling problems.

Originality and significance: I realize now that my comments under Key Results focused on originality and significance...which I stand by because I think the subject matter of the paper is far more impactful in the long run than any particular result. The authors set the stage by noting that most of the (vast) body of literature about patterns of speciation and extinction within the hominin clade has focused on the role of climate-driven environmental change (Court Jester) (lines 13-16). The focus in this paper on species interactions as a driver of macroevolutionary dynamics (Red Queen) within the hominin clade is novel, as is the approach for studying macroevolutionary dynamics (at least within paleoanthropology), which borrows heavily from a body of theory that will be familiar to paleobiologists and evolutionary biologists. This study is the only one I'm aware of that explores whether speciation rates in the hominin clade are diversity-dependent and thus likely indicative of competition, and it should be of interest to paleoanthropologists, archaeologists, paleobiologists, and evolutionary biologists.

Data & methodology: Please see comments under Validity.

Appropriate use of statistics and treatment of uncertainties: Please see comments under Validity.

Conclusions: For reasons discussed under Validity, I am not yet convinced that the conclusions are robust. I could be convinced if the right steps are taken.

Suggested improvements: I would like to see a version of this paper published, but the authors will first have to modify your analytical approach to incorporate methods that are robust to the incomplete and unevenly sampled fossil record. Plenty of studies have had to contend with such problems (e.g., <https://doi.org/10.1098/rspb.2012.1793>; <https://doi.org/10.1126/science.1210695>; <https://doi.org/10.1093/sysbio/syu006>), and there are plenty of ways to go about it.

References: References are appropriate

Clarity and context: The paper is well-written.

Inflammatory material: None

Minor edits:

Line 56: 'trait' to 'traits'

The terms 'niche' and 'niche space' are used many times throughout the manuscript. Because there are so many ways to define the term, I think it's important that you be clear what you mean by 'niche' early on in the manuscript.

5Line 37: "Similar arguments have been made..." can you cite the relevant work?

Line 163: Could you briefly outline what this metric is capturing? I had to turn to the original paper to fully understand what it does.

Line 168: missing a verb somewhere (e.g., we calculated)

Line 298-299: "These patterns are unsurprising, largely because diversity-dependent speciation should operate within, and not across, adaptive grades..." This statement is rather surprising since the paragraph discussing your predictions (lines 103-111) mention nothing about adaptive grades. And then later in the discussion (lines 306-308) you decide that you would never expect the pattern in hominins but only within clades that occupy a particular adaptive grade, which seems to be at odds with the setup to the paper. Also, does the literature about density-dependent speciation imply that the pattern should only be observed within adaptive grades? I don't think so, but could be off the mark.

Line 330-338: Here's another reason to be cautious about the observation of no signal that isolated species produced more daughter-species than non-isolated species: Sampling effort also has an effect on classification of species as 'isolated' versus 'non-isolated'. Increased sampling will tend increase the observed number of species within a given locality, and eventually shift 'isolated' species into the 'non-isolated' category. It is easy to imagine, for example, that further sampling of the ~2.5 Ma fossil deposits that have yielded *A. garhi* could turn up other hominin species (ditto for *Paranthropus aethiopicus*, *P. robustus*, etc.).

Line 365-368: "Although the 95% confidence intervals of the simulated trait distributions encompass the space from 9 to 1 throughout this period, the observed jump to high disparity from ~2.8-2.25 MYA exceeds that expected under the null model, suggesting the increase in niche overlap is diffuse, but present" Reads to me like you are reading too much into a noisy dataset. If the empirical observation is within the null model expectations, then I'd be much more cautious about reading anything into the empirical observation.

Line 389: Though *P. boisei* from Malawi consumed C3 foods (Lüdecke et al., 2018)

Line 395: remove the 'at' in "...suggest at considerable..."

Line 527-528: I'd give credit to Tony Barnosky (Barnosky, 2001), since I'm pretty sure Benton borrowed the idea and figure from him.

Barnosky, A.D., 2001. Distinguishing the effects of the red queen and court jester on Miocene mammal evolution in the northern Rocky Mountains. *Journal of Vertebrate Paleontology* 21, 172-185.
Faith, J.T., Du, A., Behrensmeyer, A.K., Davies, B., Patterson, D.B., Rowan, J., Wood, B., 2021. Rethinking the ecological drivers of hominin evolution. *Trends in Ecology & Evolution*.
Lüdecke, T., Kullmer, O., Wacker, U., Sandrock, O., Fiebig, J., Schrenk, F., Mulch, A., 2018. Dietary versatility of Early Pleistocene hominins. *Proceedings of the National Academy of Sciences of the USA*

115, 13330-13335.

Maxwell, S.J., Hopley, P.J., Upchurch, P., Soligo, C., 2018. Sporadic sampling, not climatic forcing, drives early hominin diversity. *Proceeding of the National Academy of Sciences of the USA* 115, 4891-4896.

Reviewer #3 (Remarks to the Author):

This study aims to explore the potential effect of diversity dependence on speciation dynamics in hominins based on phylogenetic comparative methods. The authors estimate a negative diversity dependence in *Australopithecus* and *Paranthropus* and a positive effect in *Homo*, which they describe as evidence that the genus *Homo* evolved under fundamentally different dynamics compared to other vertebrate clades.

While the questions asked in this paper are certainly interesting, I think the methods and data used in this paper cannot adequately answer them. There are also conceptual issues in how the results are interpreted, which I think make the overall results unconvincing.

First of all, the data set is inevitably small, because hominin is a small clade. This is an issue because estimating speciation rates (which is already a contentious task e.g. doi: 10.1038/s41586-020-2176-1) does require larger trees to provide reliable estimates. A phylogeny with a dozen tips almost certainly lacks the statistical power to estimate diversity dependent diversification and rate variation through time and across clades. If it doesn't the authors should demonstrate that through simulations, since -to my knowledge- all methodological papers assessing the performance of phylogenetic birth-death models use larger trees.

Another question is how sensitive these analyses are to species sampling. The fossil record is incomplete, and chances are not all hominin species are preserved and/or have been found and described yet. What could be the effect of this incompleteness on the results?

Since the full tree with extinct and extant species is available it is not clear why the authors did not use models that incorporate this type of data recurring instead to models designed for phylogenies of extant species only (such as the DR statistic). There are several methods that can use fossil and extant lineages directly as input to estimate speciation rates e.g. in software like RevBayes, PyRate, BEAST. These methods would also more explicitly account for fossil sampling biases.

Lines 210-211: I am not sure this is how disparity plots are usually interpreted. As far as I understand they show deviations from the expectation under a Brownian motion and they always start at 1 and end at 0. Figure 5 shows that the observed patterns do not deviate significantly from a Brownian motion.

What is the confidence interval around the parameter values reported in Table 1? Given the small sample size of the data, confidence intervals around these estimates might not differ from 0 in all sub-

7clades.

Assuming that genus *Homo* indeed shows an increasing rate of speciation, why would that make the genus different from other vertebrates? There are many other explanations for the pattern, that could equally apply to other clades. For example, *Homo* could be benefitting from the diversity decline in other hominins. Or it could be the driver of such decline, by outcompeting them. *Homo* also expanded in different continents and in doing so might have expanded its niche. The expectation for a clade with expanding niche is in fact a positive relationship between standing diversity and speciation rate, just like the one found in this study. Finally, abiotic factors -neglected here- could still be controlling these patterns.

I think it is inaccurate to state that negative diversity dependence is found in “virtually all vertebrate clades”. There are several studies that do not find diversity dependence in vertebrates, e.g. turtles and crocodiles (doi: 10.1186/s12862-020-01642-3), horses (doi: 10.1126/sciadv.abc2365), primates (doi: 10.1093/sysbio/syy046) and the debate is still open as to whether a carrying capacity is even to be expected (doi: 10.1086/680859).

Author Rebuttal to Initial comments

Responses to Reviewers

Reviewer comments are pasted below, and our responses written in **bold**.

Summary of responses:

The key point raised by all Reviewers was the potential for false positive results, originating from (i) the inherently low sample size we are restricted to with the hominin fossil record, (ii) incomplete sampling, and/or (iii) sampling bias. To address these concerns, we have submitted a major rewrite of the manuscript. In short, we have removed all analyses which inherently relied upon reduced datasets (i.e., those across a smaller phylogeny, those pertaining to regional scales, and that about trait evolution), performed tests of our original methods, and, crucially, performed an extensive set of new analyses on new data in a validated Bayesian framework¹⁻³.

In more detail, we have:

1. Removed analyses reliant on reduced datasets.
2. Focused exclusively on diversity and speciation rate, to more thoroughly engage with this topic and introduce a new set of analyses. This entailed removing the analyses about trait evolution, which inherently deal with competition at a different scale than the one most of the manuscript is focused on.
3. Pooled non-*Homo* species together in our FAD-based and phylogeny-based analyses, to increase sample size for this group (where before, we had three groupings – *Australopithecus*, *Paranthropus*, and *Homo*).
4. Performed complementary analyses, based on fossil occurrence data rather than published first appearance dates (FADs) or speciation rate calculated across phylogenies, in a validated Bayesian framework¹⁻³ recommended by Reviewers 2 and 3. This method allowed us to explicitly include time- and lineage-based variability in fossil preservation rates, *and* incomplete sampling in our analyses.
5. Performed additional analyses on published FADs in the Bayesian framework described above, to estimate the effect of variability in fossil preservation rates and incomplete sampling.
6. Performed simulations to explore the performance of the phylogeny-based approach, by (i) calculating the percentage of false positive results obtained across simulated non-diversity-dependent phylogenies, (ii) calculating the percentage of false negative results obtained across simulated diversity-dependent phylogenies.
7. Performed simulations to explore the potential effect of lower sampling probability in older taxa on our finding of a difference between younger *Homo* and the older group of non-*Homo* taxa in our phylogenetic analyses.

We would like to thank the Reviewers for their patience as we revised our manuscript this year—a process delayed substantially by a total of eight months of fieldwork between the two authors!

Reviewer 1

This ms explores whether hominins have experienced diversity-dependent diversification dynamics which is found in many other (vertebrate) clades. They find evidence for negative diversity-dependence in some clades, but positive diversity-dependence in *Homo*. While I think the question is interesting and the reported result is surprising, I have concerns about the methodology, and hence doubts about the conclusiveness of the results. I will limit myself to the analysis of speciation rate and its dependence on diversity from phylogenetic information.

We are pleased that the Reviewer considers our questions and results interesting, and fully appreciate the points they raise about our methods. As was the case for Reviewers 2 and 3, these points mainly relate to sampling problems (that is, an inherently small dataset, unbalanced sampling effort, and time- and lineage-based preservation biases). We hope the Reviewer appreciates our revised manuscript, which, although we do not adopt the method they very constructively suggested, now explicitly takes into account these sampling problems in a new set of analyses suggested by Reviewers 2 and 3.

My general concern is that the number of data points is limited and experience in this field teaches us that we have very little power for inference. Just looking at Figure 2, I feel that we cannot draw strong conclusions. Related to that I suspect that the negative diversity-dependence may somehow be driven by the fact that these clades become extinct, while the *Homo* clade has not become extinct. I would like the authors to think about their results in this general way, i.e. if their result is correct, what is it in the data that causes this power to detect a signal?

The Reviewer raises the issue that sampling problems (including small sample size) and extinction can influence the patterns we report. To address these issues, we performed simulations to test the robustness of our method to low sample sizes, sampling and preservation bias, and extinction. We also performed a set of new analyses in a validated Bayesian framework¹⁻³ that explicitly takes into account these sampling problems.

We performed simulations to explore the potential effect of lower sampling probability in older taxa on our finding of a difference between younger *Homo* and the older group of non-*Homo* taxa in our phylogenetic analyses. To do so, we randomly added species to the non-*Homo* portion of the phylogeny and repeated our analyses. The difference between *Homo* and non-*Homo* was obtained across 88% of trees with 12.5% increased non-*Homo* species richness, 75% of trees with 25% and 37.5% increased non-*Homo* species richness, and 77% of trees with (a rather unlikely!) 50% increased non-*Homo* species richness.

We also performed simulations to explore the performance of the phylogeny-based approach, by (i) calculating the percentage of false positive results obtained across simulated non-diversity-dependent phylogenies, (ii) calculating the percentage of false negative results obtained across simulated diversity-dependent phylogenies. These trees included fully extinct trees as well as ultrametric trees, thus addressing the Reviewer's concern that extinction in one clade versus the other drove our results. The method is conservative across small datasets: across the small trees simulated under a diversity-dependent process, it correctly inferred diversity-dependence across 61% of the trees. The method falsely identified a relationship between diversity and speciation across 23% of the simulated constant-rate birth-death trees, but in this sample, *negative* diversity-dependence was inferred across nearly all (94%) trees. In other words, positive diversity dependent speciation was incorrectly inferred across only 1.3% of the total number of simulated trees. We take these results to suggest that there is reasonable, but not total, confidence that non-*Homo* species were characterised by negative diversity-dependence:

10

ESS
: is

there is a 77% chance that it was *not* a false positive result across a tree generated under a non-diversity-dependent process. Happily, there is only a 1.3% chance that *Homo*'s positive diversity-dependent speciation is a methodological artefact.

Finally, we performed a new set of complementary analyses in PyRate, which allowed us to explicitly account for incomplete sampling and lineage- and time-based variability in preservation rates.

More specifically, I have concerns about the statistical approach. The authors compute speciation rates using tip DR and then do a GLM. I wonder whether tip DR is a good measure of speciation rate, especially for speciation rate through time. In any case it is still a statistic on which further statistics is done (GLM). I think there is another (arguably better) alternative which calculates the likelihood of the tree, under a model of diversity-dependent speciation (and extinction). This is quite simple for a complete tree, i.e. a tree with all extinct branches in it. I believe that the methods the authors use now also seem to make the assumption that we have complete information. Suppose we have a tree with branching or extinction events at subsequent times t_i , where $i = 1, 2, 3, \dots$ and t_0 is the stem age of the tree. We also introduce $\delta(t_i)$ as follows: if event i is a speciation event then $\delta(t_i) = 1$, otherwise $\delta(t_i) = 0$. Let us assume that the speciation rate and extinction rate depend on the number of species at time t_i through some functions $\lambda(n_i)$ and $\mu(n_i)$, respectively, where n_i is the number of species at t_{i-1} . Without any extinctions we have $n_i = i$ if we follow the phylogeny, but one can also use information on the number of species at any point in time. If there is no change in number of species between any two events in the tree, the likelihood is then given by:

$$L = \prod_{i=1}^N e^{-n_i(\lambda(n_i) + \mu(n_i))(t_i - t_{i-1})} (\lambda(n_i))^{\delta(t_i)} (\mu(n_i))^{1 - \delta(t_i)}$$

[Latex for this is: $\$L = \prod_{i=1}^N e^{-n_i(\lambda(n_i) + \mu(n_i))(t_i - t_{i-1})} (\lambda(n_i))^{\delta(t_i)} (\mu(n_i))^{1 - \delta(t_i)} \$$]

One can now use Maximum Likelihood methods or Bayesian statistics to estimate the parameters that govern the functions $\lambda(n)$ and $\mu(n)$. For instance, one can set $\lambda(n) = \max(0, \lambda_0(1 - n/K))$ and $\mu(n) = \mu_0$ as is typically done in diversity-dependent models, and parameters λ_0 , μ_0 and K are estimated. But to allow for both negative and positive diversity-dependence, one could simply take $\lambda(n) = \max(0, \lambda_0 + a n)$ for some parameter a which can be positive or negative.

If not all extinctions are observed, this can be accounted for by considering the likelihood of the fossilized birth-death model.

I would like the authors to consider this approach which is quite standard in the literature.

We very much appreciate the time and effort that clearly went in to the Reviewer's constructive suggestions. Instead of the method suggested above, however, we have adopted the additional set of analyses in PyRate requested by Reviewers 2 and 3. These analyses address the same the issues the method suggested here by the Reviewer seeks to address—namely, incomplete sampling, sampling bias, and low sample size. Of course, the analysis suggested above pertains specifically to our phylogeny-based approach, which we have kept in the manuscript—but we think the simulations to test the performance of the phylogeny-based analyses sufficiently address the concerns raised, especially in the context of our conclusions being based on results of the phylogeny *and* PyRate analyses taken together.

11

We would like to sincerely thank the Reviewer for their thoughtful and supportive suggestions, which were very helpful in directing our substantial rewrite and new analyses.

ESS
: is

Reviewer 2

Key results: This has potential to be a very important contribution because it explores a problem that has received far too little attention. A massive body of literature has examined the role of climate-driven environmental change in shaping macroevolutionary trends within fossil hominins (Court Jester hypotheses). The present paper diverges from this literature by shifting the focus to biotic interactions (Red Queen hypothesis), specifically intraspecific competition, which is typically invoked to explain density-dependent speciation patterns various taxonomic groups. This paper presents an important and long overdue counterpoint to the numerous Court Jester papers in paleoanthropology, and I could easily see a version of it becoming a key citation within the vast literature about the drivers of macroevolutionary change within the hominin clade. It is likely to open the door to new avenues of research. Likewise, the observation that *Homo* shows a unique pattern (speciation rates increase with diversity) compared to other hominin genera is going to capture people's attention.

We are delighted that the Reviewer is enthusiastic about the potential scientific impact of our paper and its implications for the field.

Validity: As much as I appreciate what the authors set out to accomplish, I have major concerns about how the incomplete and unevenly sampled fossil record has influenced several analyses that are central to the paper. Dealing with this will require that the authors rethink their analytical approach, which could be quite challenging...or fun depending on how you look at it!

We agree with the latter!

In its current form, which does not attempt to control for sampling problems, I'm not convinced that many of the results are telling us about macroevolutionary dynamics.

The need to account for sampling problems (including the inherently small sample size when dealing with the hominin clade, incomplete sampling, and temporal/geographical sampling bias) is the main, and completely justified, concern raised by the Reviewer. We agree with their point, and have substantially revised the manuscript by including new analyses based on novel data, performing simulations to test the validity of our existing results, removing analyses with inherently smaller sample sizes (those across the reduced phylogeny, those at regional scales, and those asking whether there is a difference between isolated and non-isolated species), and re-drafting the Introduction and Discussion sections.

(1) The authors begin by asking whether hominin first appearance dates (FADs) are predicted by hominin diversity in preceding time bins, the expectation being that periods of lower diversity should be followed by a greater number of FADs. This is tricky because sampling effort has a strong effect on hominin diversity (Maxwell et al., 2018) and because incomplete sampling means observed FADs are too young (Faith et al., 2021). This is true for any fossil taxon, but especially hominins because of their rarity in the fossil record. The authors demonstrate some awareness of the problem (line 91-93) but proceed with the analysis without doing anything about it. It is important that the authors acknowledge the complications inherent to the data they are working with by drawing from methodological approaches that circumvent or at least ameliorate sampling problems associated with the fossil record.

Note: The expectation that periods of lower diversity should be followed by a greater number of FADs is also consistent with the effects of sampling. Maxwell et al. (2018) show a general trend of increasing sampling effort in Africa since 7 Ma (though it drops off after 2 Ma). This trend is important since it means that, on average, FADs will tend to be preceded (500 kyr earlier) by a period of lower sampling effort and thus lower richness.

12

ESS
: is

The Reviewer rightfully makes the case that inferences of negative diversity-dependence based on the distribution of FADs by diversity may be false positives, because of increased sampling effort & preservation towards the present. We now explicitly acknowledge this (lines 276-277) in the revised manuscript.

In addition, we have taken the advice of this Reviewer and that of Reviewer 3, and (1) performed complementary analyses in PyRate¹⁻³ that estimate FADs, LADs, and speciation rates based on fossil occurrence data and explicitly taking into account temporal and lineage-based variability in fossil preservation rates; (2) applied the same analyses to published FAD and LAD dates (upon which the analysis the Reviewer correctly comments on here are based) to ask whether results differ to those using newly estimated FADs and LADs that take into account sampling issues.

We are happy to report that both (i) positive diversity-dependent speciation in *Homo*, and (ii) the differences we found between the diversity-speciation relationships of *Homo* versus non-*Homo*, were robust to incomplete sampling and sampling bias. Positive diversity-dependence in *Homo* was inferred in all Bayesian fossil occurrence with incomplete sampling + variability in preservation rates analyses.

(2) An additional analysis examines whether variation in speciation rates correlate with previous clade-wide and regional diversity. Once again, the problems associated with an incomplete fossil record complicate this analysis. We know that diversity in a given time bin (either clade-wide or at regional scales) is going to be influenced by sampling (Maxwell et al. 2018)—greater sampling effort is going to increase the number of known hominin taxa, and it will also expand their observed temporal ranges and thus increase the number of species within a given time bin. I'm less familiar with the DR metric used here (for speciation rates), but my understanding is that DR is highly sensitive to incomplete sampling and missing taxa (see here: 10.1002/ajb2.1479). Thus, I'm not sure whether correlations between speciation rates and diversity are telling us about macroevolutionary dynamics since both variables are very sensitive to sampling problems.

The Reviewer raises a similar problem for the phylogeny-based analyses, namely that sampling problems can influence the patterns we report. To address this specific issue, we performed simulations to explore the potential effect of lower sampling probability in older taxa on our finding of a difference between younger *Homo* and the older group of non-*Homo* taxa in our phylogenetic analyses. To do so, we randomly added species to the non-*Homo* portion of the phylogeny and repeated our analyses. The difference between *Homo* and non-*Homo* was obtained across 88% of trees with 12.5% increased non-*Homo* species richness, 75% of trees with 25% and 37.5% increased non-*Homo* species richness, and 77% of trees with (a rather unlikely!) 50% increased non-*Homo* species richness.

In addition, we performed simulations to explore the performance of the phylogeny-based approach, by (i) calculating the percentage of false positive results obtained across simulated non-diversity-dependent phylogenies, (ii) calculating the percentage of false negative results obtained across simulated diversity-dependent phylogenies. The method is conservative across small datasets: across the small trees simulated under a diversity-dependent process, it correctly inferred diversity-dependence across 61% of the trees. The method falsely identified a relationship between diversity and speciation across 23% of the simulated constant-rate birth-death trees, but in this sample, *negative* diversity-dependence was inferred across nearly all (94%) trees. In other words, positive diversity dependent speciation was incorrectly inferred across only 1.3% of the total number of simulated trees. We take these results to suggest that there is reasonable, but not total, confidence that non-*Homo* species were characterised by negative diversity-dependence: there is a 77% chance that it was *not* a

13

ESS
: is

false positive result across a tree generated under a non-diversity-dependent process. Happily, there is only a 1.3% chance that *Homo*'s positive diversity-dependent speciation is a methodological artefact.

We also performed a new set of complementary analyses in PyRate, which allowed us to explicitly account for incomplete sampling and lineage- and time-based variability in preservation rates, as described above.

In sum, then, we have explicitly acknowledged the sampling issues in the FAD and LAD-based analyses, performed simulations to test to what degree increased sampling effort and preservation probability towards the present affected the results of the phylogeny-based analyses, and performed additional analyses with different data (i.e., fossil occurrence data) in an analytical framework that explicitly takes into account sampling issues. We conservatively interpret the mixed results and uncertainty around inferences of negative diversity-dependence in the non-*Homo* sample as indicating no strong evidence for diversity-dependent speciation in non-*Homo*, and the consistent signals of positive diversity-dependence in *Homo*, including in models in which sampling problems are explicitly accounted for, as strong evidence for positive diversity-dependence in this genus.

Originality and significance: I realize now that my comments under Key Results focused on originality and significance...which I stand by because I think the subject matter of the paper is far more impactful in the long run than any particular result. The authors set the stage by noting that most of the (vast) body of literature about patterns of speciation and extinction within the hominin clade has focused on the role of climate-driven environmental change (Court Jester) (lines 13-16). The focus in this paper on species interactions as a driver of macroevolutionary dynamics (Red Queen) within the hominin clade is novel, as is the approach for studying macroevolutionary dynamics (at least within paleoanthropology), which borrows heavily from a body of theory that will be familiar to paleobiologists and evolutionary biologists. This study is the only one I'm aware of that explores whether speciation rates in the hominin clade are diversity-dependent and thus likely indicative of competition, and it should be of interest to paleoanthropologists, archaeologists, paleobiologists, and evolutionary biologists.

Data & methodology: Please see comments under Validity.

Appropriate use of statistics and treatment of uncertainties: Please see comments under Validity.

Conclusions: For reasons discussed under Validity, I am not yet convinced that the conclusions are robust. I could be convinced if the right steps are taken.

Suggested improvements: I would like to see a version of this paper published, but the authors will first have to modify your analytical approach to incorporate methods that are robust to the incomplete and unevenly sampled fossil record. Plenty of studies have had to contend with such problems (e.g., <https://doi.org/10.1098/rspb.2012.1793>; <https://doi.org/10.1126/science.1210695>; <https://doi.org/10.1093/sysbio/syu006>), and there are plenty of ways to go about it.

Thank you for these constructive suggestions—we have taken these on board and have added PyRate analyses as suggested.

14

References: References are appropriate

ESS
: IS

Clarity and context: The paper is well-written.

Inflammatory material: None

As we have substantially revised the manuscript, the lines/minor edits mentioned by the Reviewer that are no longer in the new manuscript are marked by an asterisk below.

Minor edits:

*Line 56: 'trait' to "traits'

The terms 'niche' and 'niche space' are used many times throughout the manuscript. Because there are so many ways to define the term, I think it's important that you be clear what you mean by 'niche' early on in the manuscript.

This is a good point, and we have now defined niche in lines 50-53.

*Line 37: "Similar arguments have been made..." can you cite the relevant work?

Line 163: Could you briefly outline what this metric is capturing? I had to turn to the original paper to fully understand what it does.

We have now added a sentence in line 128 to clarify this measure.

*Line 168: missing a verb somewhere (e.g., we calculated)

Line 298-299: "These patterns are unsurprising, largely because diversity-dependent speciation should operate within, and not across, adaptive grades..." This statement is rather surprising since the paragraph discussing your predictions (lines 103-111) mention nothing about adaptive grades. And then later in the discussion (lines 306-308) you decide that you would never expect the pattern in hominins but only within clades that occupy a particular adaptive grade, which seems to be at odds with the setup to the paper. Also, does the literature about density-dependent speciation imply that the pattern should only be observed within adaptive grades? I don't think so, but could be off the mark.

In our rewrite of the Introduction, we have clarified our approach, and justified our prediction that diversity-dependence should occur within adaptive grades (lines 92-105).

* Line 330-338: Here's another reason to be cautious about the observation of no signal that isolated species produced more daughter-species than non-isolated species: Sampling effort also has an effect on classification of species as 'isolated' versus 'non-isolated'. Increased sampling will tend increase the observed number of species within a given locality, and eventually shift 'isolated' species into the 'non-isolated' category. It is easy to imagine, for example, that further sampling of the ~2.5 Ma fossil deposits that have yielded *A. garhi* could turn up other hominin species (ditto for *Paranthropus aethiopicus*, *P. robustus*, etc.).

* Line 365-368: "Although the 95% confidence intervals of the simulated trait distributions encompass the space from 9 to 1 throughout this period, the observed jump to high disparity from ~2.8-2.25 MYA exceeds that expected under the null model, suggesting the increase in niche overlap is diffuse, but present" Reads to me like you are reading too much into a noisy dataset. If the empirical observation is within the null model expectations, then I'd be much more cautious about reading anything into the empirical observation.

15

ESS
: IS

* Line 389: Though *P. boisei* from Malawi consumed C3 foods (Lüdecke et al., 2018)

* Line 395: remove the 'at' in "...suggest at considerable..."

* Line 527-528: I'd give credit to Tony Barnosky (Barnosky, 2001), since I'm pretty sure Benton borrowed the idea and figure from him.

Barnosky, A.D., 2001. Distinguishing the effects of the red queen and court jester on Miocene mammal evolution in the northern Rocky Mountains. *Journal of Vertebrate Paleontology* 21, 172-185.

Faith, J.T., Du, A., Behrensmeyer, A.K., Davies, B., Patterson, D.B., Rowan, J., Wood, B., 2021. Rethinking the ecological drivers of hominin evolution. *Trends in Ecology & Evolution*.

Lüdecke, T., Kullmer, O., Wacker, U., Sandrock, O., Fiebig, J., Schrenk, F., Mulch, A., 2018. Dietary versatility of Early Pleistocene hominins. *Proceedings of the National Academy of Sciences of the USA* 115, 13330-13335.

Maxwell, S.J., Hopley, P.J., Upchurch, P., Soligo, C., 2018. Sporadic sampling, not climatic forcing, drives early hominin diversity. *Proceeding of the National Academy of Sciences of the USA* 115, 4891-4896.

We very much appreciated your enthusiastic, constructive, and thoughtful comments on our paper, which we feel have improved our confidence in our conclusions – thank you!

Reviewer 3

This study aims to explore the potential effect of diversity dependence on speciation dynamics in hominins based on phylogenetic comparative methods. The authors estimate a negative diversity dependence in Australopithecus and Paranthropus and a positive effect in Homo, which they describe as evidence that the genus Homo evolved under fundamentally different dynamics compared to other vertebrate clades.

While the questions asked in this paper are certainly interesting, I think the methods and data used in this paper cannot adequately answer them. There are also conceptual issues in how the results are interpreted, which I think make the overall results unconvincing.

We are pleased that the Reviewer considers our questions of interest, and appreciate the points they raise about our methods. As was the case for Reviewers 1 and 2, these points mainly relate to sampling problems—an inherently small dataset, time- and lineage-based preservation biases, and uneven sampling effort. We hope the Reviewer appreciates our revised manuscript, which now explicitly takes into account these sampling problems in a new set of analyses.

First of all, the data set is inevitably small, because hominin is a small clade. This is an issue because estimating speciation rates (which is already a contentious task e.g. doi: 10.1038/s41586-020-2176-1) does require larger trees to provide reliable estimates. A phylogeny with a dozen tips almost certainly lacks the statistical power to estimate diversity dependent diversification and rate variation through time and across clades. If it doesn't the authors should demonstrate that through simulations, since -to my knowledge- all methodological papers assessing the performance of phylogenetic birth-death models use larger trees.

The Reviewer rightfully raises the (inevitable) problem of small sample size, which limits confidence in results. To address this problem, we have performed simulations to explore the performance of the phylogeny-based approach, by (i) calculating the percentage of false positive results obtained across simulated non-diversity-dependent phylogenies, (ii) calculating the percentage of false negative results obtained across simulated diversity-dependent phylogenies. The method is conservative across small datasets: across the small trees simulated under a diversity-dependent process, it correctly inferred diversity-dependence across 61% of the trees. The method falsely identified a relationship between diversity and speciation across 23% of the simulated constant-rate birth-death trees, but in this sample, *negative* diversity-dependence was inferred across nearly all (94%) trees. In other words, positive diversity dependent speciation was incorrectly inferred across only 1.3% of the total number of simulated trees. We take these results to suggest that there is reasonable, but not total, confidence that non-*Homo* species were characterised by negative diversity-dependence: there is a 77% chance that it was *not* a false positive result across a tree generated under a non-diversity-dependent process. Happily, there is only a 1.3% chance that *Homo*'s positive diversity-dependent speciation is a methodological artefact.

We now also pair these analyses with new analyses in PyRate, as suggested below by the Reviewer, and make our inferences about processes governing hominin speciation on all results taken together.

Another question is how sensitive these analyses are to species sampling. The fossil record is incomplete, and chances are not all hominin species are preserved and/or have been found and described yet. What could be the effect of this incompleteness on the results?

17

ESS
: is

We agree with the Reviewer that incomplete sampling (and other sampling issues) can be a major problem. As Reviewer 2 also points out, there is a particular problem to be expected in the analyses contrasting non-*Homo* with *Homo* due to likely increased fossil preservation in younger lineages (i.e., *Homo*). We therefore randomly added species to the non-*Homo* portion of the phylogeny and repeated our analyses. **The difference between *Homo* and non-*Homo* was obtained across 88% of trees with 12.5% increased non-*Homo* species richness, 75% of trees with 25% and 37.5% increased non-*Homo* species richness, and 77% of trees with (a rather unlikely!) 50% increased non-*Homo* species richness.**

In addition, we have explicitly taken sampling issues into account in our complementary PyRate analyses.

Since the full tree with extinct and extant species is available it is not clear why the authors did not use models that incorporate this type of data recurring instead to models designed for phylogenies of extant species only (such as the DR statistic). There are several methods that can use fossil and extant lineages directly as input to estimate speciation rates e.g. in software like RevBayes, PyRate, BEAST. These methods would also more explicitly account for fossil sampling biases.

We are very grateful for your constructive suggestions here, and have, as suggested, performed analyses in PyRate.

Lines 210-211: I am not sure this is how disparity plots are usually interpreted. As far as I understand they show deviations from the expectation under a Brownian motion and they always start at 1 and end at 0. Figure 5 shows that the observed patterns do not deviate significantly from a Brownian motion.

We have removed these analyses in the revised manuscript, so this comment (though justified) is no longer an issue. We did so after considering, firstly, that these analyses are concerned with competition at a smaller temporal scale than that at which it is expected to affect speciation rates, and secondly, that a more constrained analytical focus would free up space to perform the additional simulations and PyRate analyses, and would be clearer for the reader.

What is the confidence interval around the parameter values reported in Table 1? Given the small sample size of the data, confidence intervals around these estimates might not differ from 0 in all sub-clades. **Table 1 has been removed from the manuscript. We now report confidence intervals for all terms in our regressions in the Supplementary Materials.**

Assuming that genus *Homo* indeed shows an increasing rate of speciation, why would that make the genus different from other vertebrates? There are many other explanations for the pattern, that could equally apply to other clades. For example, *Homo* could be benefitting from the diversity decline in other hominins. Or it could be the driver of such decline, by outcompeting them. *Homo* also expanded in different continents and in doing so might have expanded its niche. The expectation for a clade with expanding niche is in fact a positive relationship between standing diversity and speciation rate, just like the one found in this study. Finally, abiotic factors -neglected here- could still be controlling these patterns.

I think it is inaccurate to state that negative diversity dependence is found in “virtually all vertebrate clades”. There are several studies that do not find diversity dependence in vertebrates, e.g. turtles and crocodiles (doi: 10.1186/s12862-020-01642-3), horses (doi: 10.1126/sciadv.abc2365), primates (doi: 10.1093/sysbio/syy046) and the debate is still open as to whether a carrying capacity is even to be expected (doi: 10.1086/680859).

18

ESS
: is

We have taken the above to statements on board for a complete re-write of our Introduction and Discussion sections. Specifically, we explicitly engage with the Reviewer's alternative explanations for the pattern we report for *Homo* in lines 340-348. We have also emphasized the open question of ecological limits in lines 83-90, and comment on how our results fit into this general debate in lines 284-290 and 311-319.

Many thanks for your thoughtful comments and constructive suggestions – we appreciated them, and feel the conclusions we draw in the revised manuscript are more robust on their basis. Thank you!

References:

¹Silvestro, D., Salamin, N., & Schnitzler, J. (2014). PyRate: A new program to estimate speciation and extinction rates from incomplete fossil data. *Methods in Ecology and Evolution*, 5(10). <https://doi.org/10.1111/2041-210X.12263>

²Lehtonen, S., Silvestro, D., Karger, D. N., Scotese, C., Tuomisto, H., Kessler, M., Peña, C., Wahlberg, N., & Antonelli, A. (2017). Environmentally driven extinction and opportunistic origination explain fern diversification patterns. *Scientific Reports*, 7(1). <https://doi.org/10.1038/s41598-017-05263-7>

³Silvestro, D., Salamin, N., Antonelli, A., & Meyer, X. (2019). Improved estimation of macroevolutionary rates from fossil data using a Bayesian framework. *Paleobiology*, 45(4). <https://doi.org/10.1017/pab.2019.23>

Decision Letter, first revision:

20th October 2023

Dear Laura,

Your manuscript entitled "Positive diversity-dependent speciation in the genus Homo" has now been seen by the same three reviewers, whose comments are attached. The reviewers have raised a number of concerns which will need to be addressed before we can offer publication in Nature Ecology & Evolution. We will therefore need to see your responses to the criticisms raised and to some editorial concerns, along with a revised manuscript, before we can reach a final decision regarding publication.

The reviewers are pleased with the progress made in the manuscript, but still have some technical comments that will require resolution--these seem doable, so I would suggest a short/standard revision to tackle them.

We therefore invite you to revise your manuscript taking into account all reviewer and editor comments. Please highlight all changes in the manuscript text file [OPTIONAL: in Microsoft Word format].

* If you have not done so already please begin to revise your manuscript so that it conforms to our Article format instructions at <http://www.nature.com/natecolevol/info/final-submission>. Refer also to any guidelines provided in this letter.

20[REDACTED]

Nature Ecology & Evolution is committed to improving transparency in authorship. As part of our efforts in this direction, we are now requesting that all authors identified as 'corresponding author' on published papers create and link their Open Researcher and Contributor Identifier (ORCID) with their account on the Manuscript Tracking System (MTS), prior to acceptance. ORCID helps the scientific community achieve unambiguous attribution of all scholarly contributions. You can create and link your ORCID from the home page of the MTS by clicking on 'Modify my Springer Nature account'. For more information please visit please visit www.springernature.com/orcid.

[REDACTED]

Reviewer expertise:

as before

Reviewers' comments:

Reviewer #1 (Remarks to the Author):

The authors seem to have addressed many of the previous comments. Particularly the simulation study is a nice addition. However, it is not entirely clear what type of data was used from these simulations. Was it the extant-species only tree, or the full tree including extinct species? Was incomplete sampling of the tree (due to incomplete preservation) also taken into account in these simulated trees (this is done for the empirical tree but as far as I can tell not for the simulated ones)? That is, did the process mimic the actual data, including its uncertainties, as much as possible? Furthermore, I have some concerns about the analysis of the two trees separately, as it suggests that there is no interaction between lineages in either clade. This should be tested first. Ideally, a method

21similar to the one developed for extant-species phylogenies by Etienne et al 2012 (Am Nat) should be used where decoupling of diversity-dependence can be tested. This method would then also yield estimates of the patterns of diversity-dependence in either clade. I would like the authors to at least discuss such a method (I can see that developing this is not trivial). Finally, I would like to see a bit more of the data. For example, what do the trees look like?

Reviewer #2 (Remarks to the Author):

I appreciate the efforts to do revise the manuscript in light of my feedback, as well as the input from the other reviewers. I am now more convinced that key observation pertaining to positive diversity-dependent speciation in Homo is not likely driven by sampling—which was my major hang-up last time around.

A few data concerns and other suggestions for further improvement:

(1) I see that the hominin fossil occurrences were taken from the PBDB. The hominin occurrences in PBDB are notoriously spotty for Africa and even worse in Eurasia. (An informal survey of a few colleagues/collaborators -- “How good is PBDB for the hominin fossil record?” – prompted a lot of expletives) This makes me rather hesitant to accept any hominin data from PBDB as being a reasonable and unbiased reflection of the state of the hominin fossil record. Perhaps take a look at the NOW database or the ROCEEH database, and see if you can supplement some of the gaps in PBDB. Also, please include the fossil occurrence data that you used in the supplement (apologies if this was there and I just couldn’t find it)

(2) Table 1 shows that there are 7 taxa for which estimated FADs and LADs (using data from PBDB) are vastly different from the empirically observed FADs and LADs (data from Wood & Boyle). This is discussed briefly in the text on lines 295-305, but I’m still worried. For example, Homo sapiens has an estimated FAD of ~3.5 Ma, which is totally absurd (the current FAD is ~300 ka at Jebel Irhoud). How is this happening? Are there H. sapiens fossils in the database with poorly constrained ages (e.g., assigned broad Pleistocene ages of 2.588 Ma to 11.7 ka) that are somehow pushing the estimated FAD into the Pliocene? Similarly, how does Ardipithecus kadabba have an estimated extinction age of ~8.5 Ma, which is well before the species even appears? I reckon this problem merits a bit more discussion, and likely a double-checking of the data and analyses to ensure that something strange isn’t going on. The revised analyses hinge on this, so it’s important to get it right.

Other fixes:

Line 35-36: “Similar arguments have been made...” can you provide citations?

Line 61-62: “...evolutionary dispersals...” do you mean geographic dispersals? What is an evolutionary dispersal?

Line 114: “...based on FADs and LADs” – you don’t mention LADs in the subsequent text. Should this be removed from the section header?

Figure 1 A: The caption text for panel A described results for 3 genera (Homo, Paranthropus, Australopithecus) yet the graph shown in panel A shows results only for Homo and non-Homo. Please ensure that the text matches the figure.

22Figure 1 C: Perhaps I'm mis-reading this, but the correlation for Homo / Within lifetime preservation variability is colored red, indicating a significant SW (>0.5)—yet the value reported is SW: = 0.246. Shouldn't this bar be blue rather than red?

Reviewer #3 (Remarks to the Author):

I commend the authors for the thorough revisions dealing with the concerns raised in the first reviews. I think the manuscript is not more robust and focused. I still have a couple of remarks however, which I hope you will find useful.

One striking aspect of describing positive diversity-dependent effects in Homo is: how did we then end up with just one species? Positive diversity-dependent would predict a more than exponential diversity growth, that was clearly suppressed by high extinction rates. I think this question is likely to puzzle a reader from the beginning and should be discussed early in the ms. There is growing evidence that humans drove to extinctions many species, possibly including members of our own genus. I recommend including extinction in the picture because speciation can inevitably only explain one side of the evolutionary history of a clade. The fossil analyses should provide with estimates of diversity dependent extinction.

I was interested to see the new phylogenetic simulations and the analysis of the fossil data. However, if I understand correctly the MBD models were run with just one predictor (diversity), in which case a univariate model is more appropriate (https://github.com/dsilvestro/PyRate/blob/master/tutorials/pyrate_tutorial_2.md#correlation-with-a-time-continuous-variable) as it will have a simpler parameterization and should also converge a lot faster. The model also allows to use the posterior distribution of the correlation parameters (the equivalent of the G parameters shown here) to calculate the posterior probability that the effect be negative or positive, i.e. without relying on the approximation used by the shrinkage weight.

One weakness that I think remains in the analysis is that the focus is on diversity dependence while alternative explanations are not statistically tested. I am not necessarily implying that this is the case, but could e.g. global cooling equally well explain the speciation pattern? Or perhaps did increasingly open habitats favor one clade over the other? It might be worth considering these aspects in discussing the results presented here.

Line 29: I would rephrase to “diverge from other vertebrates” (I am not sure there exists a “vertebrate norm”).

Table 1: Please include the 95% credible intervals around the estimated FADs and LADs.

l. 342: a lot of literature about clade replacement and displacement could be cited here. In particular the work of Blaire Van Valkenburgh on replacement in carnivore clades is worth citing here.

*****END*****

Author Rebuttal, first revision:

Summary

We would once again like to sincerely thank the Reviewers for their time and thoughtful feedback on our last round of revisions. Beyond minor requests for changes in word choice, references, and details, the most important changes to the manuscript are:

Methods:

- Highlighted in green in the manuscript file:
The creation of a new fossil occurrence database incorporating data the Paleobiology Database (as in the previous version of the manuscript), but also (1) the NOW database, (2) ROCEEH ROAD database, and (3) manual additions based on a literature review. The addition of occurrence data from the NOW and ROCEEH databases was requested by Reviewer 2. As a result, we now have a database with balanced and accurate coverage across the world—this was a justified concern of the Reviewer, because the Paleobiology Database had poor coverage outside of Africa. This new database alone will, we imagine, be of wide interest to the field. We then ran the Bayesian models incorporating the same fossil preservation variability priors as in the previous version of the manuscript on two levels of occurrence specificity (two operational definitions of localities (at the finest-grained occurrence level available [n=486 occurrences], and at the broadest occurrence level [i.e., in which all occurrences at a site complex were merged into a single occurrence; n=355 occurrences])).
- Highlighted in orange in the manuscript file:
A broadening of the scope of the manuscript to include the relationship between diversity and extinction, which was requested by Reviewer 3. The Bayesian birth-death models we ran on the fossil occurrence data automatically output the correlation between diversity and extinction, as the Reviewer pointed out; and we agreed that this fits within the remit of our previous questions, but strengthens the manuscript in that we can now give a full account of the effect of diversity on hominin macroevolution.

Results/interpretation

24- Highlighted in blue in the manuscript file:
With the updated fossil occurrence database, we now have a much stronger signal of negative diversity-dependent speciation across the clade as a whole than in the previous version of the manuscript, and have updated our interpretation of the results to reflect this. Paired with the weak signal of negative diversity-dependent speciation in the non-*Homo* subclade (same result as in previous manuscript), we interpret this result to reflect the possibility of clade-wide diversity limits that governed speciation overall, but which were not quite reached by the non-*Homo* subclade before their extinction.

Reviewer #1:

The authors seem to have addressed many of the previous comments. Particularly the simulation study is a nice addition. However, it is not entirely clear what type of data was used from these simulations. Was it the extant-species only tree, or the full tree including extinct species?

We are very pleased the Reviewer is happy with our previous set of revisions!

We used trees with extinct species to most closely approximate the real phylogeny, and we have updated the manuscript to specify this in line 222.

Was incomplete sampling of the tree (due to incomplete preservation) also taken into account in these simulated trees (this is done for the empirical tree but as far as I can tell not for the simulated ones)? That is, did the process mimic the actual data, including its uncertainties, as much as possible?

We didn't do this in the previous round of revisions, as we saw our two simulation studies as serving two distinct functions (testing the rates of Type I and II errors, and asking how sensitive the results were to sampling bias)—but were happy to include a third simulation in which we removed, at random, tips from the simulated trees. The results are described in lines 298-311.

Furthermore, I have some concerns about the analysis of the two trees separately, as it suggests that there is no interaction between lineages in either clade. This should be tested first. Ideally, a method similar to the one developed for extant-species phylogenies by Etienne et al 2012 (Am Nat) should be used where decoupling of diversity-dependence can be tested. This method would then also yield estimates of the patterns of diversity-dependence in either clade. I would like the authors to at least discuss such a method (I can see that developing this is not trivial).

We assume the Reviewer means that we analysed subtrees comprising *Homo* and a subtree comprising non-*Homo* separately, and we see why this would be a problem, but this is not the case: to clarify, we calculated DR for all tips on a single phylogeny, and explicitly compared the two groups' tip DR in a phylogenetic GLS (again using the tree with all species on to account for phylogenetic signal in the residual error). We used overall diversity (i.e., *Homo* and non-*Homo*) as predictor, because this is also what was done in the Bayesian models based on fossil occurrence data. Apologies if we misunderstood the Reviewer's comment; we would be happy to address it differently if this was the

case!

Finally, I would like to see a bit more of the data. For example, what do the trees look like?

We have included the phylogeny, with tips coloured to indicate to which group (*Homo* or non-*Homo*) species belong, in Figure 1b.

Thank you very much for taking the time to read through our revised manuscript, and for providing your second set of comments, which we hope we have sufficiently addressed.Reviewer #2:

I appreciate the efforts to do revise the manuscript in light of my feedback, as well as the input from the other reviewers. I am now more convinced that key observation pertaining to positive diversity-dependent speciation in Homo is not likely driven by sampling—which was my major hang-up last time around.

A few data concerns and other suggestions for further improvement:

We are very pleased the Reviewer appreciated the steps we took to address their concerns in our previous rewrite of the manuscript, and that they now consider our results more robust. We have fully taken on board and implemented the Reviewer’s suggestion to supplement the PBDB data with the NOW and ROCEEH databases (details below), and have further confidence in the patterns we report as a result of doing so. Many thanks for your constructive feedback!

(1) I see that the hominin fossil occurrences were taken from the PBDB. The hominin occurrences in PBDB are notoriously spotty for Africa and even worse in Eurasia. (An informal survey of a few colleagues/collaborators -- “How good is PBDB for the hominin fossil record?” – prompted a lot of expletives) This makes me rather hesitant to accept any hominin data from PBDB as being a reasonable and unbiased reflection of the state of the hominin fossil record. Perhaps take a look at the NOW database or the ROCEEH database, and see if you can supplement some of the gaps in PBDB. Also, please include the fossil occurrence data that you used in the supplement (apologies if this was there and I just couldn’t find it)

We supplemented the PBDB occurrence data with all hominin occurrence entries from the NOW and ROCEEH databases. We manually identified duplicates and retained record with most up-to-date age estimates. If occurrences did not have a specified accession number, we identified duplicates based on location (geological formation and/or member, latitude and longitude) in combination with inspecting specified source publications (if available). To account for differences in the three databases’ approach to defining occurrence localities (e.g., *Australopithecus afarensis* at Laetoli comprises two entries in the Paleobiology Database, both of which are composites of >2 find spots, whilst all find spots are separate entries in the NOW database), we took a hierarchical approach to recording occurrences, recording ‘Site complex’ (e.g., the Woranso-Mille paleoanthropological

28research area), 'Site' (e.g., Taung), 'Subsite' (e.g., localities or surface find spots within a 'site'; subsite 'type' was also recorded), 'Formation' (e.g., Koobi Fora), 'Stratigraphic unit' (e.g., Member 4). Not all occurrences have information for all variables: for example, the Mauer site is not part of a larger 'Site complex'. We supplemented and updated the merged database with occurrence information obtained from literature reviews of papers published after 2016, and cross-checked our database with occurrence information supplied in published overviews of research where available. From these data, we created four new sets of estimated times of speciation and extinction with two sets of explicit fossil preservation rate priors and two operational definitions of localities (at the finest-grained occurrence level available [n=486 occurrences], and at the broadest occurrence level [i.e., in which all occurrences at a site complex were merged into a single occurrence; n=355 occurrences]).

(2) Table 1 shows that there are 7 taxa for which estimated FADs and LADs (using data from PBDB) are vastly different from the empirically observed FADs and LADs (data from Wood & Boyle). This is discussed briefly in the text on lines 295-305, but I'm still worried. For example, *Homo sapiens* has an estimated FAD of ~3.5 Ma, which is totally absurd (the current FAD is ~300 ka at Jebel Irhoud). How is this happening? Are there *H. sapiens* fossils in the database with poorly constrained ages (e.g., assigned broad Pleistocene ages of 2.588 Ma to 11.7 ka) that are somehow pushing the estimated FAD into the Pliocene? Similarly, how does *Ardipithecus kadabba* have an estimated extinction age of ~8.5 Ma, which is well before the species even appears? I reckon this problem merits a bit more discussion, and likely a double-checking of the data and analyses to ensure that something strange isn't going on. The revised analyses hinge on this, so it's important to get it right.

The newly estimated FADs and LADs are now much closer in line with the Wood & Boyle dates, although they do lengthen species' lifespans. The models estimated that species originated, on average 0.49 million years earlier (within-lifetime variability preservation prior) and 0.37 million years earlier (time-based variability preservation prior) than published dates suggest, and went extinct 0.27 million years later (within-lifetime variability) and 0.15 million years later (time-based variability) than published dates suggest. This is, however, what is expected given that the priors account for uncertainty in FAD and LAD estimation generated by fossil preservation variability; and, interestingly, brings hominin species lifespans closer to the mammalian average of 1 million years in many cases. We discuss this in lines 425-442.

Other fixes:

Line 35-36: "Similar arguments have been made..." can you provide citations?

We have cut this line from the MS entirely to make space for more details elsewhere.

Line 61-62: “...evolutionary dispersals...” do you mean geographic dispersals? What is an evolutionary dispersal?

On reflection, this was a blunder in diction, and we have changed “evolutionary” to “geographic”.

Line 114: “...based on FADs and LADs” – you don’t mention LADs in the subsequent text. Should this be removed from the section header?

We have removed this analysis so this comment has not been implemented.

Figure 1 A: The caption text for panel A described results for 3 genera (Homo, Paranthropus, Australopithecus) yet the graph shown in panel A shows results only for Homo and non-Homo.

This figure has been updated—it is now Figure 2—and we have re-written the caption text.

Figure 1 C: Perhaps I’m mis-reading this, but the correlation for Homo / Within lifetime preservation variability is colored red, indicating a significant SW (>0.5)—yet the value reported is SW: = 0.246. Shouldn’t this bar be blue rather than red?

We have removed this analysis so this comment has not been implemented.

We very much appreciated constructive comments and would particularly like to thank you for drawing our attention to the other two databases, the inclusion of which has improved our confidence in our conclusions – thank you!

Reviewer #3:

I commend the authors for the thorough revisions dealing with the concerns raised in the first reviews. I think the manuscript is not more robust and focused. I still have a couple of remarks however, which I hope you will find useful.

We are very happy the Reviewer is positive about our previous round of changes and we hope, of course, for the same outcome about the way we addressed their current set of very helpful suggestions.

One striking aspect of describing positive diversity-dependent effects in Homo is: how did we then end up with just one species? Positive diversity-dependent would predict a more than exponential diversity growth, that was clearly suppressed by high extinction rates. I think this question is likely to puzzle a reader from the beginning and should be discussed early in the ms. There is growing evidence that humans drove to extinctions many species, possibly including members of our own genus. I recommend including extinction in the picture because speciation can inevitably only explain one side of the evolutionary history of a clade. The fossil analyses should provide with estimates of diversity dependent extinction.

We have fully implemented this suggestion and have shifted the manuscript's focus to include extinction. All sections pertaining to extinction are indicated with orange text in the manuscript file.

I was interested to see the new phylogenetic simulations and the analysis of the fossil data. However, if I understand correctly the MBD models were run with just one predictor (diversity), in which case a univariate model is more appropriate (https://github.com/dsilvestro/PyRate/blob/master/tutorials/pyrate_tutorial_2.md#correlation-with-a-time-continuous-variable) as it will have a simpler parameterization and should also converge a lot faster. The model also allows to use the posterior distribution of the correlation parameters (the equivalent of the G parameters shown here) to calculate the posterior probability that the effect be negative or positive, i.e. without relying on the approximation used by the shrinkage weight.

We have fully implemented this suggestion and ran birth-death models with a time continuous variable (i.e., diversity).

One weakness that I think remains in the analysis is that the focus is on diversity dependence while

31alternative explanations are not statistically tested. I am not necessarily implying that this is the case, but could e.g. global cooling equally well explain the speciation pattern? Or perhaps did increasingly open habitats favor one clade over the other? It might be worth considering these aspects in discussing the results presented here.

We appreciate the point the Reviewer is making here, especially in light of the heavy emphasis on climate in human evolutionary studies. In response, we have:

- **Clarified that we are simply asking whether patterns reported (without explicit contrasts with climate!) in other clades is also present in hominins (lines 105-108, 448-451)**
- **Explicitly addressed climate as a confounding variable in the Abstract and Discussion (lines 30, 411-416)**
- **Explicitly addressed that climate and competition likely operated together in human evolution, and that the historic emphasis on one necessitates, as a first step, an explicit focus on the other, but also that our results raise new questions about the interaction between climate and competition (lines 446-456).**

Line 29: I would rephrase to “diverge from other vertebrates” (I am not sure there exists a “vertebrate norm”).

We have rewritten this line to now read “Exploring how hominin macroevolution fits into the general vertebrate macroevolutionary landscape has the potential to offer new perspectives on longstanding questions in vertebrate evolution, and shed new light on evolutionary processes within our own lineage.” (lines 33-36)

Table 1: Please include the 95% credible intervals around the estimated FADs and LADs.

We have included the 95% CIs around these in the manuscript (Table 1), and those for the other newly generated sets of FADs and LADS in the supplementary materials.

l. 342: a lot of literature about clade replacement and displacement could be cited here. In particular the work of Blaire Van Valkenburgh on replacement in carnivore clades is worth citing here.

We have included this in the current manuscript (line 404)

We would like to sincerely thank the Reviewer for their thoughtful and supportive suggestions, which were very helpful in speeding up our analyses and have made our manuscript more comprehensive.

Decision Letter, second revision:

12th February 2024

Dear Laura,

Thank you for submitting your revised manuscript "Unusual and unexpected diversity-dependent speciation and extinction in Homo" (NATECOLEVOL-220716968B). It has now been seen again by the original reviewers and their comments are below. The reviewers find that the paper has improved in revision, and therefore we'll be happy in principle to publish it in Nature Ecology & Evolution, pending minor revisions to satisfy the reviewers' final requests and to comply with our editorial and formatting guidelines.

[REDACTED]

Reviewer #2 (Remarks to the Author):

The authors have done a great job responding to our feedback. This has been a productive and rewarding process and I reckon this paper has potential to get people thinking about the drivers of hominin evolution in exciting new ways. I have nothing left to add except for one tiny little thing.

I had previously noted:

"Line 61-62: "...evolutionary dispersals..." do you mean geographic dispersals? What is an evolutionary dispersal?"

To which the response was:

33"On reflection, this was a blunder in diction, and we have changed "evolutionary" to "geographic"."

Note that the text still says "evolutionary dispersal" rather than "geographic dispersal"

Reviewer #3 (Remarks to the Author):

The authors did an excellent job revising their study. I only have few remaining suggestions for (minor) edits:

Line 21: replace 'asking' with 'testing'?

Line 66: it *is* unknown

Line 192: NHPP = *non*-homogenous Poisson process

Line 202: it might be better replacing 'exponential birth-death process' with something like: 'birth-death model in which the speciation and extinction rates are determined by an exponential correlation to a time-variable predictor'. This would be less confusing especially since all birth-death processes are 'exponential' in that they assume exponentially distributed waiting times.

Line 362: I would change to 'only for few groups, including island dwelling beetles [...]'

Line 418": replace 'There are two alternative explanations' with 'we propose two [...]'?

Line 436: Note that in a birth-death process the duration of a species is only determined by extinction rates, not by speciation rate.

Our ref: NATECOLEVOL-220716968B

28th February 2024

34Dear Dr. van Holstein,

Thank you for your patience as we've prepared the guidelines for final submission of your Nature Ecology & Evolution manuscript, "Unusual and unexpected diversity-dependent speciation and extinction in Homo" (NATECOLEVOL-220716968B). Please carefully follow the step-by-step instructions provided in the attached file, and add a response in each row of the table to indicate the changes that you have made. Please also check and comment on any additional marked-up edits we have proposed within the text. Ensuring that each point is addressed will help to ensure that your revised manuscript can be swiftly handed over to our production team.

****We would like to start working on your revised paper, with all of the requested files and forms, as soon as possible (preferably within two weeks). Please get in contact with us immediately if you anticipate it taking more than two weeks to submit these revised files.****

In recognition of the time and expertise our reviewers provide to Nature Ecology & Evolution's editorial process, we would like to formally acknowledge their contribution to the external peer review of your manuscript entitled "Unusual and unexpected diversity-dependent speciation and extinction in Homo". For those reviewers who give their assent, we will be publishing their names alongside the published article.

Nature Ecology & Evolution offers a Transparent Peer Review option for new original research manuscripts submitted after December 1st, 2019. As part of this initiative, we encourage our authors to support increased transparency into the peer review process by agreeing to have the reviewer comments, author rebuttal letters, and editorial decision letters published as a Supplementary item. When you submit your final files please clearly state in your cover letter whether or not you would like to participate in this initiative. Please note that failure to state your preference will result in delays in accepting your manuscript for publication.

Cover suggestions

We welcome submissions of artwork for consideration for our cover. For more information, please see our guide for cover artwork.

Please submit your suggestions, clearly labeled, along with your final files. We'll be in touch if more

35information is needed.

Nature Ecology & Evolution has now transitioned to a unified Rights Collection system which will allow our Author Services team to quickly and easily collect the rights and permissions required to publish your work. Approximately 10 days after your paper is formally accepted, you will receive an email in providing you with a link to complete the grant of rights. If your paper is eligible for Open Access, our Author Services team will also be in touch regarding any additional information that may be required to arrange payment for your article.

Please note that *Nature Ecology & Evolution* is a Transformative Journal (TJ). Authors may publish their research with us through the traditional subscription access route or make their paper immediately open access through payment of an article-processing charge (APC). Authors will not be required to make a final decision about access to their article until it has been accepted. Find out more about Transformative Journals

Authors may need to take specific actions to achieve compliance with funder and institutional open access mandates. If your research is supported by a funder that requires immediate open access (e.g. according to Plan S principles) then you should select the gold OA route, and we will direct you to the compliant route where possible. For authors selecting the subscription publication route, the journal's standard licensing terms will need to be accepted, including <https://www.nature.com/nature-portfolio/editorial-policies/self-archiving-and-license-to-publish>. Those licensing terms will supersede any other terms that the author or any third party may assert apply to any version of the manuscript.

[REDACTED]

[REDACTED]

Reviewer #2:

Remarks to the Author:

The authors have done a great job responding to our feedback. This has been a productive and rewarding process and I reckon this paper has potential to get people thinking about the drivers of hominin evolution in exciting new ways. I have nothing left to add except for one tiny little thing.

36I had previously noted:

"Line 61-62: "...evolutionary dispersals..." do you mean geographic dispersals? What is an evolutionary dispersal?"

To which the response was:

"On reflection, this was a blunder in diction, and we have changed "evolutionary" to "geographic"."

Note that the text still says "evolutionary dispersal" rather than "geographic dispersal"

Reviewer #3:

Remarks to the Author:

The authors did an excellent job revising their study. I only have few remaining suggestions for (minor) edits:

Line 21: replace 'asking' with 'testing'?

Line 66: it *is* unknown

Line 192: NHPP = *non*-homogenous Poisson process

Line 202: it might be better replacing 'exponential birth-death process' with something like: 'birth-death model in which the speciation and extinction rates are determined by an exponential correlation to a time-variable predictor'. This would be less confusing especially since all birth-death processes are 'exponential' in that they assume exponentially distributed waiting times.

Line 362: I would change to 'only for few groups, including island dwelling beetles [...]'

Line 418": replace 'There are two alternative explanations' with 'we propose two [...]'?

Line 436: Note that in a birth-death process the duration of a species is only determined by extinction rates, not by speciation rate.

Author Rebuttal, second revision:

POINT-BY-POINT RESPONSES TO REVIEWERS

We would like to sincerely thank all Reviewers for a truly enjoyable, rewarding, and thought-provoking review process!

Reviewer #2:

The authors have done a great job responding to our feedback. This has been a productive and rewarding process and I reckon this paper has potential to get people thinking about the drivers of hominin evolution in exciting new ways. I have nothing left to add except for one tiny little thing.

I had previously noted:

"Line 61-62: "...evolutionary dispersals..." do you mean geographic dispersals? What is an evolutionary dispersal?"

To which the response was:

"On reflection, this was a blunder in diction, and we have changed "evolutionary" to "geographic"."

Note that the text still says "evolutionary dispersal" rather than "geographic dispersal"

A second blunder! We've now double-checked this, and our resubmitted manuscript says "geographic dispersal" (line 65-66).

Reviewer #3:

The authors did an excellent job revising their study. I only have few remaining suggestions for (minor) edits:

Line 21: replace 'asking' with 'testing'?

We have implemented this change (line 21).

Line 66: it *is* unknown

We have implemented this change (line 66).

Line 192: NHPP = *non*-homogenous Poisson process

We have implemented this change (line 444).

Line 202: it might be better replacing 'exponential birth-death process' with something like: 'birth-death model in which the speciation and extinction rates are determined by an exponential correlation to a time-variable predictor'. This would be less confusing especially since all birth-death processes are 'exponential' in that they assume exponentially distributed waiting times.

We have changed this phrasing to: "We then applied a PyRate birth-death model in which Homo and non-Homo speciation and extinction rates were determined by an exponential correlation to a time-variable predictor, in this case clade-wide lineage-through-time estimates for each set of times of origination and extinction." (lines 453-456)

Line 362: I would change to 'only for few groups, including island dwelling beetles [...]'

We have implemented this change (lines 266-267).

Line 418": replace 'There are two alternative explanations' with 'we propose two [...]'?

We have implemented this change (line 322).

Line 436: Note that in a birth-death process the duration of a species is only determined by extinction rates, not by speciation rate.

We have deleted the line this is referring to, to avoid confusion.

Final Decision Letter:

8th March 2024

Dear Laura,

We are pleased to inform you that your Article entitled "Diversity-dependent speciation and extinction in hominins", has now been accepted for publication in *Nature Ecology & Evolution*.

Over the next few weeks, your paper will be copyedited to ensure that it conforms to *Nature Ecology and Evolution* style. Once your paper is typeset, you will receive an email with a link to choose the appropriate publishing options for your paper and our Author Services team will be in touch regarding any additional information that may be required

Due to the importance of these deadlines, we ask you please us know now whether you will be difficult to contact over the next month. If this is the case, we ask you provide us with the contact information (email, phone and fax) of someone who will be able to check the proofs on your behalf, and who will be available to address any last-minute problems. Once your paper has been scheduled for online publication, the Nature press office will be in touch to confirm the details.

Acceptance of your manuscript is conditional on all authors' agreement with our publication policies (see www.nature.com/authors/policies/index.html). In particular your manuscript must not be published elsewhere and there must be no announcement of the work to any media outlet until the publication date (the day on which it is uploaded onto our web site).

Please note that *Nature Ecology & Evolution* is a Transformative Journal (TJ). Authors may publish their research with us through the traditional subscription access route or make their paper immediately open access through payment of an article-processing charge (APC). Authors will not be required to make a final decision about access to their article until it has been accepted. Find out more about Transformative Journals

Authors may need to take specific actions to achieve compliance with funder and institutional open access mandates. If your research is supported by a funder that requires immediate open access (e.g. according to Plan S principles) then you should select the gold OA route, and we will direct you to the compliant route where possible. For authors selecting the subscription publication route, the journal's standard licensing terms will need to be accepted, including [a href="https://www.nature.com/nature-portfolio/editorial-policies/self-archiving-and-license-to-publish"](https://www.nature.com/nature-portfolio/editorial-policies/self-archiving-and-license-to-publish). Those licensing terms will supersede any other terms that the author or any third party may assert apply to any version of the manuscript.

In approximately 10 business days you will receive an email with a link to choose the appropriate publishing options for your paper and our Author Services team will be in touch regarding any

40additional information that may be required.

We welcome the submission of potential cover material (including a short caption of around 40 words) related to your manuscript; suggestions should be sent to Nature Ecology & Evolution as electronic files (the image should be 300 dpi at 210 x 297 mm in either TIFF or JPEG format). Please note that such pictures should be selected more for their aesthetic appeal than for their scientific content, and that colour images work better than black and white or grayscale images. Please do not try to design a cover with the Nature Ecology & Evolution logo etc., and please do not submit composites of images related to your work. I am sure you will understand that we cannot make any promise as to whether any of your suggestions might be selected for the cover of the journal.

You can generate the link yourself when you receive your article DOI by entering it here: <http://authors.springernature.com/share>.

Yours sincerely,

[REDACTED]

P.S. Click on the following link if you would like to recommend Nature Ecology & Evolution to your librarian <http://www.nature.com/subscriptions/recommend.html#forms>

** Visit the Springer Nature Editorial and Publishing website at www.springernature.com/editorial-and-publishing-jobs for more information about our career opportunities. If you have any questions please click here.**